# VowelPrompt: Hearing Speech Emotions from Text via Vowel-level Prosodic Augmentation

**Yancheng Wang**[1,2*]   **Osama Hanna**[1]   **Ruiming Xie**[1]   **Xianfeng Rui**[1]   **Maohao Shen**[1,3*]
**Xuedong Zhang**[1]   **Christian Fuegen**[1]   **Jilong Wu**[1]   **Debjyoti Paul**[1]   **Arthur Guo**[1]
**Zhihong Lei**[1]   **Ozlem Kalinli**[1]   **Qing He**[1]   **Yingzhen Yang**[2]
[1]Meta Superintelligence Labs, [2]Arizona State University, [3]Massachusetts Institute of Technology
yancheng.wang@asu.edu   ohanna@meta.com

## Abstract

Emotion recognition in speech presents a complex multimodal challenge, requiring comprehension of both linguistic content and vocal expressivity, particularly prosodic features such as fundamental frequency, intensity, and temporal dynamics. Although large language models (LLMs) have shown promise in reasoning over textual transcriptions for emotion recognition, they typically neglect fine-grained prosodic information, limiting their effectiveness and interpretability. In this work, we propose VowelPrompt, a linguistically grounded framework that augments LLM-based emotion recognition with interpretable, fine-grained vowel-level prosodic cues. Drawing on phonetic evidence that vowels serve as primary carriers of affective prosody, VowelPrompt extracts pitch-, energy-, and duration-based descriptors from time-aligned vowel segments, and converts these features into natural language descriptions for better interpretability. Such a design enables LLMs to jointly reason over lexical semantics and fine-grained prosodic variation. Moreover, we adopt a two-stage adaptation procedure comprising supervised fine-tuning (SFT) followed by Reinforcement Learning with Verifiable Reward (RLVR), implemented via Group Relative Policy Optimization (GRPO), to enhance reasoning capability, enforce structured output adherence, and improve generalization across domains and speaker variations. Extensive evaluations across diverse benchmark datasets demonstrate that VowelPrompt consistently outperforms state-of-the-art emotion recognition methods under zero-shot, fine-tuned, cross-domain, and cross-linguistic conditions, while enabling the generation of interpretable explanations that are jointly grounded in contextual semantics and fine-grained prosodic structure.

## 1 Introduction

Paralinguistic speech understanding requires modeling not only what is said but how it is said with prosodic patterns in fundamental frequency ($F_0$), intensity (RMS energy), timing (duration, rhythm, pause), and voice quality. Speech emotion recognition (SER) is commonly framed either with discrete categories, such as, angry, sad, happy, neutral, or with dimensional labels in the valence–arousal–dominance space, and evaluated on acted and naturalistic corpora (Busso et al., 2008; Poria et al., 2019; Cao et al., 2014; Livingstone & Russo, 2018; Russell, 1980; Bradley & Lang, 1994). Classic SER pipelines extract engineered low-level descriptors (LLDs) and functionals via OPENSMILE (Eyben et al., 2010) and standardized sets (GeMAPS/eGeMAPS) (Eyben et al., 2015), chosen specifically for interpretability in paralinguistics. Recent advances (Pepino et al., 2021; Yang et al., 2021) are driven by self-supervised speech representation learning methods, such as wav2vec 2.0 (Chen et al., 2022), HuBERT (Hsu et al., 2021), and WavLM (Chen et al., 2022), which provide robust utterance representations and often set strong baselines on SER and SUPERB-style evaluations. While effective, these embeddings are opaque and require an audio encoder at inference time, which complicates interpretability and deployment in text-only settings.

---

*Work done during internship at Meta Superintelligence Labs.

Large language models (LLMs) have introduced two complementary paths for spoken affect. Audio language LLMs such as AudioPaLM (Rubenstein et al., 2023), SALMONN (Tang et al., 2024), Qwen2-Audio (Chu et al., 2024), and task-specific Emotion-LLaMA (Cheng et al., 2024), integrate continuous acoustic encoders with LLM backbones to reason directly over speech (Rubenstein et al., 2023). In parallel, text-only prompting augments ASR transcripts with natural-language descriptions of prosody (e.g., "spoken loudly with rising intonation"), enabling LLMs to exploit affective cues without consuming raw audio (Wu et al., 2025), motivated by the strong reasoning capabilities of text LLM DeepSeek-AI et al. (2025); Wang et al. (2024); Zhao et al. (2025); Wei et al. (2022). The latter is lightweight and interpretable but typically uses coarse, utterance-level descriptors that can blur fine-grained cues.

Phonetic evidence suggests that phoneme classes contribute unequally to affective cues. Vowels, voiced nuclei with relatively stable $F_0$ and energy, often carry salient intonation patterns; syllable nuclei have also been used to localize prosodic variation (Ringeval & Chetouani, 2008). At the same time, class-aggregated analyses indicate that consonantal regions can encode complementary or even stronger spectral evidence for emotion in some settings (Bitouk et al., 2010). This motivates a segment-centric representation that emphasizes vowel nuclei to capture fine-grained prosodic structure, while preserving the full lexical context.

We propose VowelPrompt, a simple yet effective interpretable augmentation method for LLM-based speech emotion recognition. Given an utterance and its transcript, the method first obtains time-aligned vowel segments through a standard forced-alignment pipeline. It then extracts vowel-level low-level descriptors, including $F_0$ level and slope, $F_0$ variability, intensity level and variability, and segment duration, applying both speaker and vowel-type normalization. These values are discretized via quantile binning and converted into concise natural-language prosodic descriptors such as "high $F_0$, rising, loud, lengthened." The resulting descriptors are appended to the transcript so that a text-only LLM can jointly reason over lexical content and segment-level prosody. Model adaptation follows a two-stage regimen, beginning with supervised fine-tuning (SFT) and continuing with Reinforcement Learning with Verifiable Reward (RLVR) using Group Relative Policy Optimization (GRPO) to improve reasoning quality, output-format adherence, and robustness while maintaining proximity to the SFT reference (McAuliffe et al., 2017; DeepSeek-AI et al., 2025).

**Contributions.** The contributions of this paper are summarized as follows.

First, leveraging well-established phonetic evidence, VowelPrompt extracts vowel-level prosodic descriptors, including pitch level and contour, intensity, and temporal duration, from time-aligned segments obtained via forced alignment, applies both speaker- and vowel-type normalization, and discretizes these features into natural language descriptions. These interpretable descriptors are appended to transcripts, enabling LLMs to jointly reason over lexical semantics and localized prosodic variation, in contrast to opaque acoustic embeddings.

Second, to adapt LLMs to this enriched input, we design a two-stage training pipeline that begins with supervised fine-tuning (SFT) for cold-start alignment and continues with Reinforcement Learning with Verifiable Rewards (RLVR) using Group Relative Policy Optimization (GRPO), which improves structural adherence, robustness, and reasoning quality.

Third, extensive experiments on five benchmark datasets, including IEMOCAP (Busso et al., 2008), MELD (Poria et al., 2019), CaFE (Gournay et al., 2018), EmoDB (Burkhardt et al., 2005), and ASVP-ESD (Tientcheu Touko et al., 2021), demonstrate that VowelPrompt consistently surpasses competitive baselines across zero-shot, few-shot, fine-tuned, cross-domain, and multilingual conditions, while enabling interpretable and verifiable emotion reasoning grounded in both linguistic and prosodic information.

## 2 RELATED WORKS

### 2.1 SPEECH EMOTION RECOGNITION AND PARALINGUISTIC ANALYSIS

Speech emotion recognition (SER) aims to infer a speaker's affective state from acoustic signals, often leveraging prosodic, spectral, and linguistic features. Early SER systems relied heavily on low-level descriptors such as fundamental frequency (F0), energy, and temporal statistics, extracted via toolkits like openSMILE (Eyben et al., 2010). The INTERSPEECH Computational Paralinguis-

tics Challenge series established standardized feature sets such as the Geneva Minimalistic Acoustic Parameter Set (GeMAPS) (Eyben et al., 2015), which provide interpretable descriptors covering pitch, loudness, and voice quality. Deep learning methods have since outperformed handcrafted features in performance, with wav2vec 2.0-based embeddings (Pepino et al., 2021) and contextualized transformer encoders such as EmoBERTa (Kim & Provost, 2021) achieving state-of-the-art results. However, these high-dimensional representations are difficult to interpret, making it challenging to explain or control model predictions in sensitive applications. Recent advances integrate language models with acoustic or visual modalities to improve emotion reasoning. Prompt-based augmentation has been explored, where prosodic descriptions (e.g., "spoken loudly with rising intonation") are prepended to transcripts to guide large language models (Wu et al., 2025). This approach yields measurable improvements in zero-shot emotion recognition, particularly in clean speech settings. At the architectural level, multimodal models such as AudioPaLM (Rubenstein et al., 2023) and Emotion-LLaMA (Cheng et al., 2024) fuse audio embeddings directly into transformer-based LLMs, enabling joint reasoning over text and audio inputs. While effective, these systems typically rely on audio embeddings learned through black-box models, which limit their interpretability. Our work bridges this gap by combining interpretable vowel-level acoustic features with textual prompting, enabling accuracy gains while preserving human-readable intermediate representations.

## 2.2 VOWEL-CENTRIC PROSODY IN EMOTIONAL SPEECH

Phonetic studies consistently highlight vowels as primary carriers of emotional prosody. Vowels, being voiced and acoustically stable, exhibit clear correlates of affect such as pitch level, contour, intensity, and duration (Ringeval & Chetouani, 2008). Ringeval & Chetouani (2008) has demonstrated that vowel-based acoustic features improve emotion classification compared to utterance-level statistics, while Schuller et al. (2009) has found that class-level spectral features for vowels and consonants can capture complementary emotional cues. Subsequent work in articulatory phonetics found that emotional states systematically shift vowel articulation and formant positions, influencing both perceived tone and loudness (Shah & Busso, 2019). Despite these findings, most modern SER pipelines extract features uniformly across all phonemes, potentially diluting the discriminative power of vowel-specific prosodic cues.

Vowels, as voiced phonemes characterized by a relatively open vocal tract configuration, dominate both the acoustic energy and temporal duration of spoken utterances. They convey a substantial portion of prosodic information, including pitch (fundamental frequency), intensity (perceived loudness), and temporal patterns (duration and rhythm), which are critical to paralinguistic expression and emotional communication (Crystal, 1969; Mozziconacci, 2002). Extensive phonetic research has shown that vowels function as primary carriers of intonation contours and emotional coloration, owing to their sustained voicing and spectral stability. Building on these insights, we construct a structured, interpretable intermediate representation that focuses on vowel-centric acoustic features as a bridge between raw audio signals and downstream language models.

## 3 METHODS

We propose VowelPrompt, a unified framework that enriches LLMs with interpretable vowel-level prosodic cues for enhanced speech emotion recognition. The central premise is that vowels, which carry the majority of the energy and prosodic variation of the speech signal, serve as fine-grained indicators. To exploit this property, VowelPrompt first derives discrete, human-interpretable descriptors of pitch, intensity, and temporal dynamics from individual vowel segments, isolating localized acoustic events that often correspond to emotionally salient moments in speech. These descriptors are converted into natural language and integrated directly into the input prompts alongside the textual transcript, enabling the LLM to reason jointly over lexical semantics and prosodic structure. The model is then adapted to the emotion recognition task via a two-phase training regime. The supervised fine-tuning phase aligns the LLM's generation behavior to produce accurate, well-reasoned emotion predictions conditioned on both textual and prosodic information, while the reinforcement learning phase refines the LLM's reasoning quality, adherence to output format, and robustness to speaker and context variability. This design bridges the interpretability of phonetic-level analysis with the reasoning capabilities of modern LLMs, yielding a system that can explicitly link

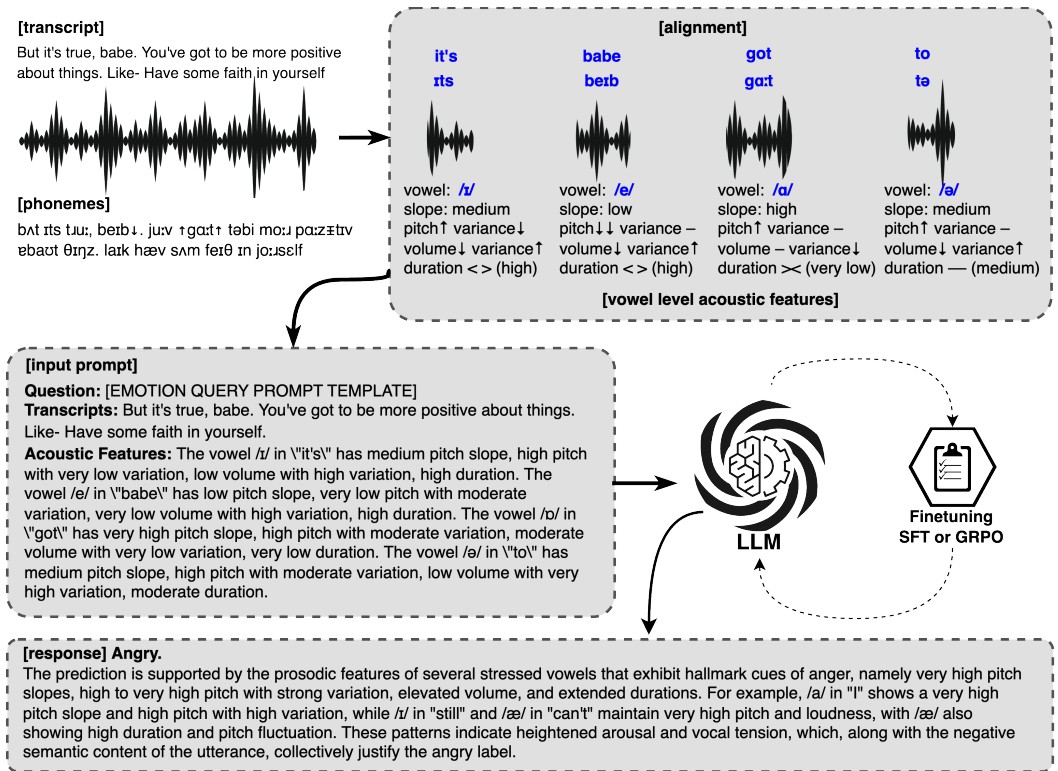

Figure 1: An example of the proposed VowelPrompt framework for the emotion recognition task.

acoustic–prosodic patterns to emotion categories in an interpretable manner. Figure 1 illustrates an example of the proposed VowelPrompt for the emotion recognition task.

## 3.1 VOWEL-LEVEL ACOUSTIC FEATURE EXTRACTION

**Forced Alignment and Vowel Selection.** Given an utterance and its orthographic transcript, we employ phoneme-level forced alignment to obtain precise temporal boundaries for each phoneme. Vowel segments are then extracted based on a predefined inventory derived from the International Phonetic Alphabet (IPA), encompassing both monophthongs and diphthongs. This selective filtering excludes consonantal segments, isolating the voiced, resonance-rich nuclei that are most informative for prosodic and affective analysis. By anchoring our vowel selection to IPA standards, we ensure cross-linguistic consistency and compatibility with multilingual phonetic analysis pipelines for languages including English, German, and French.

**Low-Level Descriptor Extraction.** For each vowel segment, we compute a compact set of low-level descriptors (LLDs) that are both human-interpretable and suitable for integration into large language models, as presented in Table 1. The LLDs used as the acoustic features include (1) **average pitch** ($F_0$) and **pitch slope**, which jointly capture the segment's intonation level and rising/falling trends; (2) **pitch variation**, defined as the within-segment standard deviation of $F_0$, indicating the degree of dynamic modulation; (3) **average intensity** and **intensity variation**, which reflect loudness and its fluctuation; and (4) **duration**, representing the temporal extent of the vowel and conveying information about speech rate and emphasis. Pitch and intensity features are computed using Praat-style signal processing algorithms (Boersma & Weenink, 2001), configured with speaker-adaptive floor and ceiling parameters to account for individual vocal characteristics, while segment durations are derived directly from the phoneme-level forced alignment boundaries.

To ensure comparability across speakers and vowel categories, we employ a two-stage normalization. First, we apply speaker-level $z$-normalization to control for individual voice characteristics. Second, vowel-type normalization is applied to mitigate systematic differences among vowel classes. The normalized continuous values are then discretized via quantile-based binning into $K$

Table 1: Vowel-level low-level descriptors (LLDs) used in VowelPrompt for prosodic augmentation.

| Category | Feature | Interpretation |
|---|---|---|
| **Pitch** | Pitch Level (Mean $F_0$)
Pitch Slope
Pitch Variation | Average fundamental frequency of the vowel
Rising or falling trend in pitch across the segment
Standard deviation of $F_0$, indicating dynamic range |
| **Intensity** | Intensity Level
Intensity Variation | Average loudness (RMS energy) of the vowel
Fluctuation in loudness during the vowel segment |
| **Temporal** | Duration | Length of the vowel segment in seconds |

**Transcripts:** Speaker_0:\"You started complaining about not being able to find a job the first day you went out and didn't find a job. I'm not saying like and not trying to you know put you down or anything, it's just that-\"\t Speaker_1:\"That's really hard to hear.\"\t Speaker_0:\"But it's true, babe. You've got to be more positive about things. Like- Have some faith in yourself.\"

**Acoustic Features**: The vowel /ɪ/ in \"it's\" has medium pitch slope, high pitch with very low variation, low volume with high variation, high duration. The vowel /e/ in \"babe\" has low pitch slope, very low pitch with moderate variation, very low volume with high variation, high duration. The vowel /ɒ/ in \"got\" has very high pitch slope, high pitch with moderate variation, moderate volume with very low variation, very low duration. The vowel /ə/ in \"to\" has medium pitch slope, high pitch with moderate variation, low volume with very high variation, moderate duration.

**Question:** Please select the emotional label of <Speaker_0:\"But it's true, babe. You've got to be more positive about things. Like- Have some faith in yourself.\"> based on both the conversation transcript and the acoustic features. Please output ONLY ONE label from <happy, sad, neutral, angry, excited, frustrated>. Return the reasoning process in <think> </think> and final emotion in <answer> </answer> tags

**Response: <think>**The repeated urging to "be more positive" and "have some faith" implies an emotional buildup and suggests that Speaker_0 is not simply offering support, but is also reacting to prolonged negativity with a degree of emotional strain. This is mirrored in the acoustic features: vowels such as /ɒ/ in "got" and /ʌ/ in "some" exhibit very high pitch slopes and variation, signaling emotional intensity and urgency; the vowel /ɒ/ in "positive" shows high pitch variation and very long duration, suggesting a strong, emotionally charged emphasis. The overall vocal pattern, marked by intensity, uneven dynamics, and deliberate stress on emotionally salient words, aligns with a frustrated emotional state.**</think> <answer>**frustrated**</answer>**

Figure 2: Example of a prompt of VowelPrompt combining conversational context, target utterance, and vowel-level prosodic descriptors. The transcript provides lexical content, while each vowel in the target utterance is annotated with interpretable acoustic features, including pitch slope, pitch level and variation, intensity level and variation, and duration. These features are expressed in natural language and integrated into the input to guide the emotion inference by LLM. The example illustrates a frustration-labeled case from IEMOCAP, where prosodic patterns such as high pitch slope and extended vowel duration convey heightened emotional intensity.

ordinal categories (e.g., "very low," "low," "moderate," "high," "very high"), with $K$ selected to balance interpretability and resolution.

**Natural Language Conversion.** The discretized features are deterministically mapped into concise textual descriptors for each vowel segment. This process is parameter-free, ensuring transparency and reproducibility. The resulting descriptors can be appended to transcripts. Compared to sentence-level acoustic summaries, vowel-level descriptors capture fine-grained, localized prosodic variation that often aligns with emotionally salient or emphasized words. This representation provides higher temporal resolution, direct interpretability for human analysts, and flexibility to serve as controllable units in expressive speech generation or style transfer.

## 3.2 FINE-TUNING LLM FOR EMOTION RECOGNITION WITH VOWEL-LEVEL ACOUSTIC FEATURES

We adopt a two-stage fine-tuning pipeline to adapt a Large Language Model (LLM) for emotion recognition using the extracted vowel-level acoustic features described in Section 3.1. The first stage, supervised fine-tuning (SFT), serves as a cold-start adaptation, while the second stage, reinforcement learning with verifiable rewards (RLVR), further refines reasoning accuracy and output structure. Figure 2 illustrates an example of VowelPrompt fine-tuned by SFT and RL for better reasoning over the context and acoustic features for emotion recognition.

**Supervised Fine-Tuning (SFT).** In the SFT stage, we augment each utterance's textual transcript with its corresponding vowel-level prosodic descriptors in natural language form, following a fixed

prompt template. This augmentation explicitly grounds the LLM in acoustic cues, enabling it to reason over both lexical semantics and prosodic dynamics. To establish a cold-start alignment with the target task, we use only a small portion of the available training data, paired with gold reasoning traces automatically generated by a high-capacity text-only LLM such as GPT-4o (Hurst et al., 2024). These reasoning traces serve as reference outputs, allowing the target LLM to learn both the correct label and an interpretable reasoning process. We initialize from a pretrained instruction-tuned LLM and fine-tune with cross-entropy loss to maximize the likelihood of generating the reference reasoning and correct emotion label.

**Reinforcement Learning with Verifiable Reward (RLVR).** Following SFT, we finetune the LLM using Reinforcement Learning with Verifiable Reward (RLVR) (DeepSeek-AI et al., 2025), which jointly optimizes reasoning accuracy and adherence to a prescribed output format. Given an input prompt $q$ containing both the transcript and its aligned prosodic feature descriptions, the policy model $\pi_\theta$ produces an output $o$ consisting of two distinct components, including an explicit reasoning trace enclosed within `<think></think>` tags, and a final predicted emotion enclosed within `<answer></answer>` tags. Such an explicit separation enables independent, rule-based verification of both the reasoning process and the final prediction.

To perform RLVR, we define a composite reward that integrates an accuracy-based term $R_{\text{acc}}$ and a format-based term $R_{\text{format}}$:

$$R(o, y) = R_{\text{acc}}(o, y) + R_{\text{format}}(o), \tag{1}$$

where $y$ denotes the ground-truth emotion label. The accuracy reward $R_{\text{acc}}$ is assigned a value of $1$ if the predicted emotion in $o$ exactly matches $y$, and $0$ otherwise. The format reward $R_{\text{format}}$ is assigned a value of $1$ if $o$ contains both a syntactically valid reasoning block (`<think>...</think>`) and a final answer block (`<answer>...</answer>`); otherwise, it is set to $0$. Both components are deterministic and require no learned parameters, ensuring the verifiability of the reward signal.

**Group Relative Policy Optimization.** We optimize the response generation policy using Group Relative Policy Optimization (GRPO), which encourages each candidate response to outperform the group average while maintaining diversity (DeepSeek-AI et al., 2025). To stabilize training and prevent drift from the supervised initialization, we add a KL penalty that constrains updates relative to the SFT reference model. This lightweight formulation enables verifiable reward optimization without requiring complex learned reward models.

## 3.3 Multilingual Extension with IPA-based Vowel Mapping

To extend VowelPrompt to multilingual SER, we adopt a language-agnostic framework grounded in the International Phonetic Alphabet (IPA) to unify vowel representations across languages. Such adaptation enables consistent extraction of vowel-level prosodic descriptors regardless of language-specific phoneme inventories or orthographic conventions.

**Phoneme Alignment and IPA Normalization.** For each language, we employ a phoneme-level forced alignment tool capable of aligning speech to phonemic transcriptions in the target language. In our experiments, we use Montreal Forced Aligner (MFA) (McAuliffe et al., 2017), which supports over 20 languages with pretrained acoustic and grapheme-to-phoneme (G2P) models. Aligned phonemes are then mapped into a shared set of IPA symbols to ensure phonetic comparability across languages. To control for cross-lingual variation in prosodic realization, we further perform normalization at the language level. For each of the languages considered in this paper, including English, German, and French, we compute global means and standard deviations for each prosodic feature and apply $z$-score normalization within that language.

**Prompt Construction and Adaptation.** Once normalized and discretized, the resulting vowel-level descriptors are converted into natural language descriptions in English. The generated acoustic features are appended to the transcript. We use multilingual LLMs, such as GPT-4o (Hurst et al., 2024) and Qwen2-7B-Instruct (Yang et al., 2024), that natively support the input language. For SFT, we finetune these models using multilingual emotion datasets, preserving the same prompt structure and training objectives as described in Section 3.2.

## 4 EXPERIMENTS

This section presents a rigorous empirical evaluation of VowelPrompt across five widely-used speech emotion recognition benchmarks under a range of experimental configurations. The dataset characteristics are summarized in Section 4.1. Section 4.2 examines zero-shot emotion recognition performance relative to existing prompting-based baselines, while Section 4.3 investigates the effectiveness of SFT and GRPO. The generalizability of VowelPrompt under domain shift is assessed in Section 4.4, and its applicability to multilingual emotion recognition is explored in Section 4.5. In the appendix, Section A.1 presents a feature-level ablation study to assess the individual contributions of vowel-level prosodic descriptors, and Section A.2 analyzes the comparative performance of zero-shot and few-shot prompting. Section A.3 provides a direct comparison between VowelPrompt and a projection-based baseline incorporating the audio embeddings for emotion recognition with LLMs. In Section A.4, we perform a study on the number of quantization bins $K$ used for discretizing continuous vowel-level acoustic features. Section A.5 analyzes the influence of utterance duration on zero-shot recognition performance in MELD. In Section A.6, we conduct a series of ablation and counterfactual experiments to verify that VowelPrompt's predictions are driven by aligned vowel-level prosodic descriptors rather than spurious lexical or formatting cues, including transcript shuffle, prosody permutation, matched-marginal placebo, and cross-swap analysis. In Section A.7, we study the impact of label verbalizer tokenization by analyzing the tokenization behavior of emotion labels under different tokenizers and by replacing the original labels with synthetic single-token verbalizers under randomized mappings. In Section A.8, we compare vowel-level prosodic descriptors with consonant-level prosodic descriptors and further evaluate a variant that incorporates both types of cues. In Section A.9, we compare VowelPrompt with non-LLM acoustic baseline methods based on self-supervised speech representations, including HuBERT-large and wav2vec-large, under both in-domain and cross-domain evaluation settings. In Section A.10, we perform a human evaluation of the reasoning traces generated by VowelPrompt, where annotators assess prosodic grounding, causal coherence, and internal consistency. In Section A.11, we analyze the balance between in-domain performance and cross-domain generalization by varying the KL regularization weight in GRPO. In Section A.12, we compare VowelPrompt with classifiers trained directly on vowel-level prosodic features, including MLP, tree-based models, and a transformer operating on phoneme-level tokens. In Section A.13, we study the impact of incorrect vowel alignment by perturbing a controlled fraction of vowel boundary timestamps in the forced alignment results. In Section A.14, we analyze the effect of speech rate by grouping utterances according to phones-per-second statistics and evaluating performance across different speech-rate regimes. In Section A.15, we perform an ablation study on the incorporation of explicit reasoning in both SFT and SFT combined with GRPO by enabling and disabling reasoning during training.

### 4.1 DATASETS

We evaluate our method on five widely used speech emotion recognition (SER) benchmarks that span acted, semi-acted, and naturalistic speech across multiple languages. The IEMOCAP corpus (Busso et al., 2008) contains dyadic interactions between ten actors (five male, five female), with utterances annotated for emotions including angry, happy, sad, neutral, and excited. The MELD dataset (Poria et al., 2019) is derived from the TV series Friends, consisting of multiparty conversations annotated with seven emotion categories in a multimodal setting. To assess cross-lingual generalization, we further include three public benchmarks, including CaFE (Gournay et al., 2018) in French, EmoDB (Burkhardt et al., 2005) in German, and the multilingual ASVP-ESD (Tientcheu Touko et al., 2021), which covers 12 emotions across diverse speakers and recording conditions. The statistics of all the datasets used are summarized in Table 2.

Table 2: Summary of emotion recognition datasets used in our experiments.

| Dataset | Source | Language | #Emotions | #Speakers | #Utterances | #Hours |
|---|---|---|---|---|---|---|
| IEMOCAP (Busso et al., 2008) | Act | English | 5 | 10 | 5531 | 7.0 |
| MELD (Poria et al., 2019) | TV | English | 7 | 304 | 13706 | 12.1 |
| CaFE (Gournay et al., 2018) | Act | French | 7 | 12 | 936 | 1.2 |
| EmoDB (Burkhardt et al., 2005) | Act | German | 7 | 10 | 535 | 0.5 |
| ASVP-ESD (Tientcheu Touko et al., 2021) | Media | Mix | 12 | 131 | 13964 | 18.0 |

## 4.2 Zero-Shot Emotion Recognition

We evaluate the proposed VowelPrompt approach in a zero-shot setting on the IEMOCAP and MELD datasets, comparing it against two baselines, including a vanilla zero-shot prompt using only transcripts, denoted as Zero-Shot Baseline, and SpeechCueLLM (Wu et al., 2025), which augments transcripts with sentence-level prosodic descriptions. For each method, we evaluate two input configurations: (i) Transcript, which utilizes solely the target utterance, and (ii) Transcript & Context, which additionally incorporates preceding conversational turns to provide discourse-level information. Performance is assessed using Unweighted Accuracy (UACC) and Weighted F1 (WF1), which respectively quantify class-balanced recognition capability and overall classification effectiveness.

Table 3: Zero-shot performance on IEMOCAP and MELD. Results are reported as Unweighted Accuracy / Weighted F1 (%). "Context" indicates inclusion of preceding conversational turns.

| Method | Input | LLM | IEMOCAP | | MELD | |
|---|---|---|---|---|---|---|
| | | | UACC | WF1 | UACC | WF1 |
| Zero-Shot Baseline | Transcript | | 43.38 | 41.03 | 61.15 | 60.92 |
| SpeechCueLLM (Wu et al., 2025) | Transcript | GPT-4o | 49.97 | 48.54 | 52.44 | 53.59 |
| **VowelPrompt (Ours)** | Transcript | | **51.18** | **50.15** | **63.61** | **61.76** |
| Zero-Shot Baseline | Transcript & Context | | 55.51 | 53.63 | 62.76 | 63.57 |
| SpeechCueLLM (Wu et al., 2025) | Transcript & Context | GPT-4o | 60.07 | 58.52 | 56.74 | 57.90 |
| **VowelPrompt (Ours)** | Transcript & Context | | **62.26** | **60.74** | **64.34** | **64.17** |
| Zero-Shot Baseline | Transcript | | 40.60 | 40.44 | 47.55 | 48.74 |
| SpeechCueLLM (Wu et al., 2025) | Transcript | LLaMA-3-8B-Instruct | 44.18 | 43.88 | 44.41 | 44.62 |
| **VowelPrompt (Ours)** | Transcript | | **46.57** | **44.96** | **49.21** | **49.99** |
| Zero-Shot Baseline | Transcript & Context | | 50.40 | 49.47 | 42.30 | 42.09 |
| SpeechCueLLM (Wu et al., 2025) | Transcript & Context | LLaMA-3-8B-Instruct | 52.63 | 53.85 | 43.49 | 42.59 |
| **VowelPrompt (Ours)** | Transcript & Context | | **53.82** | **54.10** | **46.45** | **46.26** |

As shown in Table 3, VowelPrompt consistently outperforms both baselines across models and datasets. On GPT-4o, VowelPrompt improves over the Zero-Shot Baseline by up to 7.80% UACC and 7.11% WF1 on IEMOCAP, and by up to 2.19% UACC and 3.25% WF1 on MELD. Compared to SpeechCueLLM, our method achieves gains in all settings, indicating that fine-grained vowel-level prosodic cues are more effective than coarse sentence-level descriptions for emotion recognition in large language models. The trend holds for LLaMA-3-8B-Instruct, despite its weaker overall performance compared to GPT-4o. Even in this resource-constrained LLM, VowelPrompt yields consistent improvements over both baselines, with gains of up to 3.64% UACC and 3.63% WF1. These results demonstrate that VowelPrompt is a portable, model-agnostic prompting strategy that can enhance zero-shot emotion recognition without task-specific fine-tuning.

## 4.3 LLM Fine-Tuning for Emotion Recognition

We further evaluate VowelPrompt in a supervised adaptation setting to examine whether vowel-level prosodic augmentation yields benefits beyond zero-shot prompting. Experiments are conducted on IEMOCAP and MELD with two instruction-tuned LLM backbones, which are LLaMA-3-8B-Instruct (Dubey et al., 2024) and LLaMA-4-Scout-17B-16E-Instruct (Meta AI, 2025). For the SFT setting, the reasoning is not incorporated into the training and inference processes for VowelPrompt and the baseline methdos. For the SFT & GRPO setting, both models are adapted using LoRA-based parameter-efficient fine-tuning on 20% of the training data, followed by GRPO as described in Section 3.2. We use the official train/validation/test splits for each dataset, and all methods are trained and evaluated on identical utterance–label pairs to ensure fair comparison. Similar to the settings for the zero-shot experiments, we conduct comparisons across multiple input configurations. The *Baseline* leverages only the transcript and preceding conversational turns without incorporating any prosodic information. InstructERC (Lei et al., 2023) applies instruction tuning to enhance context-sensitive emotion recognition. SALMONN (Tang et al., 2024) integrates speech and language modalities through multimodal alignment. SpeechCueLLM (Wu et al., 2025) augments the transcript with sentence-level prosodic summaries. Finally, VowelPrompt enriches the input with fine-grained, interpretable prosodic descriptors for each vowel segment, as described in Section 3.1. Each method is evaluated under both SFT and SFT & GRPO regimes, enabling a systematic as-

sessment of the benefits of prosodic granularity, multimodal integration, and reinforcement-based refinement.

Table 4: Weighted F1 (%) on IEMOCAP and MELD under SFT and SFT & GRPO settings with different LLMs.

| Method | LLaMA-3-8B-Instruct | | | | LLaMA-4-Scout-17B-16E-Instruct | | | |
| | SFT | | SFT & GRPO | | SFT | | SFT & GRPO | |
| | IEMOCAP | MELD | IEMOCAP | MELD | IEMOCAP | MELD | IEMOCAP | MELD |
|---|---|---|---|---|---|---|---|---|
| Baseline | 70.32 | 67.44 | – | – | 70.82 | 67.90 | – | – |
| InstructERC (Lei et al., 2023) | 71.65 | 67.25 | 71.32 | 66.96 | 71.75 | 68.15 | 71.52 | 67.35 |
| SALMONN (Tang et al., 2024) | 71.36 | 67.25 | 71.02 | 66.85 | 71.48 | 67.96 | 71.85 | 67.10 |
| SpeechCueLLM (Wu et al., 2025) | 71.74 | 67.07 | 71.55 | 67.10 | 72.02 | 68.02 | 72.18 | 67.96 |
| **VowelPrompt (Ours)** | **73.46** | **69.61** | **73.02** | **68.98** | **73.85** | **70.12** | **74.02** | **69.79** |

As shown in Table 4, VowelPrompt consistently outperforms all competing baselines across both datasets and model scales. Under SFT, vowel-level augmentation yields absolute Weighted F1 improvements of up to $3.14\%$ on IEMOCAP and $2.17\%$ on MELD with LLaMA-3-8B-Instruct, with comparable gains observed for the larger LLaMA-4-Scout model. The advantage remains after RLVR refinement, where VowelPrompt outperforms sentence-level prosodic descriptions by as much as $1.47\%$ on IEMOCAP and $1.88\%$ on MELD. These results demonstrate that fine-grained, interpretable vowel-centric features encode richer emotional cues than coarse prosodic summaries, and that RLVR refinement can further capitalize on these cues to improve classification performance.

## 4.4 Cross-Domain Emotion Recognition

We further assess the robustness of VowelPrompt under domain shift through cross-domain evaluations, where models are trained on one dataset and directly tested on another without additional adaptation. Specifically, we examine two transfer scenarios, which are from IEMOCAP to MELD, and from MELD to IEMOCAP. The study evaluates whether VowelPrompt can capture emotional cues that generalize across variations in speaker identity, conversational style, and recording conditions. Following the protocol in Section 4.3, we compare VowelPrompt against SpeechCueLLM (Wu et al., 2025), which augments transcripts with sentence-level prosodic descriptions. Both methods are tested under three regimes: zero-shot prompting, supervised fine-tuning (SFT), and SFT followed by GRPO (SFT & GRPO). All experiments employ the LLaMA-3-8B-Instruct backbone, with training performed on the full source-domain dataset.

Table 5: Cross-domain results for IEMOCAP → MELD and MELD → IEMOCAP. Models are trained on the source dataset and evaluated on the target dataset without adaptation.

| Method | IEMOCAP → MELD | | | MELD → IEMOCAP | | |
| | Zero-Shot | SFT | SFT & GRPO | Zero-Shot | SFT | SFT & GRPO |
|---|---|---|---|---|---|---|
| SALMONN (Tang et al., 2024) | - | 40.25 | 51.48 | - | 23.65 | 40.85 |
| InstructERC (Lei et al., 2023) | 51.42 | 43.15 | 50.18 | 42.68 | 25.49 | 43.36 |
| SpeechCueLLM (Wu et al., 2025) | 53.85 | 42.36 | 55.16 | 42.59 | 25.10 | 44.79 |
| **VowelPrompt (Ours)** | 54.10 | 46.26 | **60.28** | 46.26 | 28.71 | **51.75** |

As shown in Table 5, VowelPrompt consistently outperforms all baselines across transfer settings. Gains are modest in the zero-shot condition but increase substantially with supervised adaptation. Under SFT & GRPO, VowelPrompt improves by $5.12\%$ in the IEMOCAP → MELD transfer and by $6.96\%$ in the MELD → IEMOCAP transfer compared to SpeechCueLLM. These findings indicate that fine-grained vowel-level acoustic features provide more domain-invariant emotional cues than coarse sentence-level summaries, and that RL-based refinement further enhances cross-domain generalization.

## 4.5 Extracting Vowel-Level Acoustic Features from Multilingual Speech

To evaluate cross-lingual generalization, we extend VowelPrompt to three additional benchmarks: the French CaFE corpus (Gournay et al., 2018), the German EmoDB corpus (Burkhardt et al.,

2005), and the mixed-lingual ASVP-ESD corpus (Tientcheu Touko et al., 2021). Phoneme-level forced alignment is performed using the Montreal Forced Aligner (MFA), after which vowel segments are mapped into a shared IPA-based inventory. Prosodic features, including pitch, intensity, and duration, are normalized at both the speaker and language level before being converted into natural-language descriptors. Moreover, we conduct zero-shot evaluations on CaFE and EmoDB with GPT-4o, comparing against transcript-only baselines, InstructERC (Lei et al., 2023), and SpeechCueLLM (Wu et al., 2025). For ASVP-ESD, which is inherently multilingual, we perform supervised adaptation using Qwen2-7B-Instruct, chosen for its stronger multilingual capabilities. The evaluation compares VowelPrompt against InstructERC, SALMONN (Tang et al., 2024), and SpeechCueLLM under both SFT and SFT & GRPO training regimes.

Table 6: Zero-shot results on CaFE (French) and EmoDB (German) using GPT-4o. Performance is reported as Weighted F1 (%).

| Method | CaFE (Fr) | EmoDB (De) |
|---|---|---|
| Transcript Only | 45.10 | 64.86 |
| InstructERC (Lei et al., 2023) | 48.35 | 66.74 |
| SpeechCueLLM (Wu et al., 2025) | 49.16 | 67.32 |
| **VowelPrompt (Ours)** | **51.42** | **69.85** |

Table 7: Fine-tuning results on ASVP-ESD (Mixlingual) using Qwen2-7B-Instruct. Performance is reported as Weighted F1 (%).

| Method | SFT | SFT & GRPO |
|---|---|---|
| InstructERC (Lei et al., 2023) | 67.25 | 67.96 |
| SALMONN (Tang et al., 2024) | 67.10 | 67.85 |
| SpeechCueLLM (Wu et al., 2025) | 67.85 | 68.12 |
| **VowelPrompt (Ours)** | **70.54** | **71.36** |

As shown in Tables 6 and 7, VowelPrompt achieves consistent improvements over all baselines across languages and evaluation settings. In the zero-shot scenario, it delivers the best F1 scores on both CaFE and EmoDB, outperforming transcript-only prompts, InstructERC, and SpeechCueLLM, thereby demonstrating effective transferability without language-specific supervision. On the mixed-lingual ASVP-ESD corpus, supervised adaptation with SFT & GRPO further improves performance, where VowelPrompt outperforms InstructERC, SALMONN, and SpeechCueLLM, underscoring the effectiveness of vowel-level prosodic augmentation in multilingual contexts.

## 5 CONCLUSION

In this work, we introduced VowelPrompt, a unified and interpretable framework that augments large language models with fine-grained, vowel-level prosodic cues for speech emotion recognition. Grounded in phonetic theory, VowelPrompt extracts prosodic descriptors of pitch, intensity, and duration from time-aligned vowel segments, discretizes them through quantile-based binning, and converts them into natural language descriptions appended to transcripts. This design enables language models to reason jointly over lexical and prosodic information without requiring direct access to raw audio at inference. To enhance task adaptation, we developed a two-stage training pipeline combining supervised fine-tuning with Reinforcement Learning using Verifiable Reward (RLVR) via Group Relative Policy Optimization (GRPO), which improves predictive accuracy, structural consistency, and robustness. Comprehensive experiments across zero-shot, fine-tuned, cross-domain, and multilingual settings demonstrate that VowelPrompt consistently outperforms transcript-only and sentence-level prosody baselines. Beyond improved performance, the framework offers interpretable intermediate representations that explicitly connect acoustic–prosodic patterns to emotional categories, providing both practical effectiveness and scientific transparency for prosody-aware emotion recognition with language models.

### ACKNOWLEDGMENTS

Y. Wang and Y. Yang are supported by the 2023 Mayo Clinic and Arizona State University Alliance for Health Care Collaborative Research Seed Grant Program under Award No. AWD00038846 and by the NIH under Award No. 1OT2OD037955-01.

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

# A ADDITIONAL EXPERIMENT RESULTS

## A.1 ABLATION STUDY ON INDIVIDUAL ACOUSTIC FEATURES

To disentangle the contributions of each vowel-level descriptor, we perform a fine-grained ablation study by selectively removing one feature at a time from the six categories listed in Table 1. In particular, we evaluate the impact of excluding pitch level, pitch slope, pitch variation, intensity level, intensity variation, and duration while keeping all other descriptors intact. Each ablation model is trained under the same supervised fine-tuning (SFT) protocol with LLaMA-3-8B-Instruct on IEMOCAP and MELD to ensure comparability. This design allows us to assess the relative importance of each feature type for emotion recognition. As shown in Table 8, the removal of any single descriptor results in modest but consistent decreases in performance relative to the full model. All ablation settings preserve competitive results, with scores above 72.5% on IEMOCAP and 69.05% on MELD, confirming that VowelPrompt does not rely disproportionately on a single cue. Among the six descriptors, pitch-related features (level, slope, variation) exhibit the most noticeable impact, reflecting their well-established role as primary carriers of prosodic information. Intensity and duration features also contribute measurable improvements, as their exclusion reduces recognition accuracy despite more subtle effects. Taken together, these findings demonstrate that each vowel-level descriptor contributes complementary information to the framework, and that the integration of all six is necessary to achieve optimal performance.

Table 8: Ablation of individual vowel-level features under SFT with LLaMA-3-8B-Instruct.

| Model Variant | IEMOCAP | MELD |
|---|---|---|
| Full VowelPrompt (all features) | **73.46** | **69.61** |
| w/o Pitch Level | 72.91 | 69.18 |
| w/o Pitch Slope | 73.02 | 69.27 |
| w/o Pitch Variation | 72.87 | 69.12 |
| w/o Intensity Level | 73.15 | 69.25 |
| w/o Intensity Variation | 72.94 | 69.09 |
| w/o Duration | 73.25 | 69.22 |

## A.2 FEW-SHOT EMOTION RECOGNITION

We assess the performance of VowelPrompt in both zero-shot and few-shot scenarios on the IEMO-CAP and MELD datasets, focusing on the Transcript & Context configuration. In the few-shot

setting, each prompt is augmented with three labeled in-context exemplars drawn from the training data, enabling the models to leverage limited supervision in addition to their inherent zero-shot reasoning capability. All results are reported in terms of Weighted F1 (WF1), which provides a balanced measure of classification performance under label imbalance.

The results in Table 9 show that all methods obtain consistent improvements in the few-shot regime, with WF1 gains ranging from approximately $0.8\%$ to $1.2\%$ relative to zero-shot performance. Across both model backbones, VowelPrompt achieves the best results, outperforming the baseline and SpeechCueLLM in both evaluation settings. These findings indicate that vowel-level prosodic descriptors not only strengthen zero-shot emotion recognition but also enhance few-shot generalization, demonstrating their effectiveness as interpretable and transferable cues for prosody-aware large language models.

Table 9: Zero-shot vs. few-shot performance on IEMOCAP and MELD with Transcript & Context inputs.

| Method | IEMOCAP | | | MELD | | |
|---|---|---|---|---|---|---|
| | Zero-Shot | Few-Shot | $\Delta$ | Zero-Shot | Few-Shot | $\Delta$ |
| GPT-4o | | | | | | |
| Baseline | 53.63 | 54.42 | +0.79 | 63.57 | 64.51 | +0.94 |
| SpeechCueLLM (Wu et al., 2025) | 58.52 | 59.41 | +0.89 | 57.90 | 58.95 | +1.05 |
| VowelPrompt (Ours) | **60.74** | **61.72** | +0.98 | **64.17** | **65.20** | +1.03 |
| LLaMA-3-8B-Instruct | | | | | | |
| Baseline | 49.47 | 50.26 | +0.79 | 42.09 | 43.05 | +0.96 |
| SpeechCueLLM (Wu et al., 2025) | 53.85 | 54.71 | +0.86 | 42.59 | 43.66 | +1.07 |
| VowelPrompt (Ours) | **54.10** | **55.12** | +1.02 | **46.26** | **47.42** | +1.16 |

## A.3 Comparison with Projection-Based Audio Incorporation Method

To further assess the impact of vowel-level augmentation, we compare VowelPrompt with two ablation models, including a transcript-only baseline, where the LLM is fine-tuned on textual transcripts and conversational context without any prosodic cues, and a projection-based audio encoder baseline, where continuous acoustic embeddings from Whisper are temporally pooled and passed through a learned projection module into the LLaMA token space. Both approaches are evaluated under supervised fine-tuning (SFT) on IEMOCAP and MELD, using LLaMA-3-8B-Instruct and LLaMA-4-Scout-17B-16E-Instruct backbones. This comparison highlights the trade-offs between purely textual inputs, continuous projection-based augmentation, and discrete interpretable vowel-level descriptors.

Table 10: Comparison of transcript-only, projection-based audio encoders (Whisper + projector), and VowelPrompt (VowelPrompt) under supervised fine-tuning (SFT). Results are reported as Weighted F1 (%).

| Model | IEMOCAP | | | MELD | | |
|---|---|---|---|---|---|---|
| | Transcript-Only | Projection | VowelPrompt | Transcript-Only | Projection | VowelPrompt |
| LLaMA-3-8B-Instruct | 70.32 | 72.65 | **73.46** | 67.44 | 68.85 | **69.61** |
| LLaMA-4-Scout-17B | 70.82 | 73.05 | **73.85** | 67.90 | 69.32 | **70.12** |

As shown in Table 10, both projection-based augmentation and VowelPrompt yield clear gains over the transcript-only baseline, underscoring the value of incorporating prosodic information. Among the augmentation strategies, VowelPrompt achieves the best results across all settings, outperforming the projection-based baseline method on both IEMOCAP and MELD, and across both LLaMA-3 and LLaMA-4 backbones.

## A.4 Ablation Study on the Number of Bins $K$

The number of quantization bins $K$ used for discretizing continuous vowel-level acoustic features determines the balance between interpretability and granularity. With very small $K$ (e.g., $K = 2$),

Table 11: Ablation on the number of bins $K$ for quantile-based discretization of vowel-level features. Results are reported as Weighted F1 (%).

| $K$ | Zero-Shot | | SFT | |
|---|---|---|---|---|
| | IEMOCAP | MELD | IEMOCAP | MELD |
| 2 | 57.45 | 61.32 | 71.12 | 67.28 |
| 3 | 58.72 | 62.18 | 72.04 | 68.01 |
| 4 | 59.86 | 63.47 | 73.02 | 69.05 |
| 5 | **60.74** | **64.17** | **73.46** | **69.61** |
| 6 | 60.22 | 63.89 | 73.12 | 69.18 |
| 7 | 59.74 | 63.41 | 72.78 | 68.92 |
| 8 | 59.15 | 62.95 | 72.33 | 68.40 |

the descriptors are overly coarse and fail to capture fine prosodic variation. Increasing $K$ improves resolution, but excessively large values, such as $K \geq 7$, introduce sparsity and noisy distinctions, reducing model generalization. To assess this effect, we perform an ablation study on IEMOCAP and MELD under both zero-shot prompting and supervised fine-tuning (SFT). Results are presented in Table 11. Performance improves steadily as $K$ increases from 2 to 5, with $K = 5$ consistently achieving the best results across all datasets and training regimes. Beyond this point, performance declines slightly, indicating that excessive discretization is detrimental. These findings support $K = 5$ as the optimal setting, striking a balance between interpretability and discriminative power in VowelPrompt.

### A.5 ANALYSIS BY UTTERANCE DURATION ON MELD

To further examine how utterance duration influences model performance, we analyze zero-shot results on the MELD dataset by grouping test utterances into short ($<$1s), medium (1s–3s), and long ($>$3s) categories. Table 12 reports both Unweighted Accuracy (UACC) and Weighted F1 (WF1) scores for GPT-4o and LLaMA-3-8B-Instruct under transcript-only prompting, SpeechCueLLM (Wu et al., 2025), and VowelPrompt. As shown in Table 12, performance declines as utterances grow longer, reflecting the increased variability and contextual complexity of extended speech. Despite this trend, VowelPrompt consistently provides improvements over both baselines across all duration categories. The gains are especially pronounced for short and long utterances, where vowel-level cues help disambiguate emotions that may otherwise be blurred by brevity or diluted in extended discourse. This demonstrates that VowelPrompt remains robust across diverse temporal scales of spoken dialogue.

Table 12: Zero-shot performance on MELD under different utterance durations. Results are reported as Unweighted Accuracy / Weighted F1 (%).

| Method | LLM | Target Utterance Duration | | |
|---|---|---|---|---|
| | | $<$1s | 1s–3s | $>$3s |
| Transcript Only | | 67.03 / 66.92 | 65.17 / 64.28 | 54.34 / 55.20 |
| SpeechCueLLM (Wu et al., 2025) | GPT-4o | 59.50 / 60.39 | 55.11 / 55.96 | 47.04 / 48.84 |
| **VowelPrompt (Ours)** | | **69.53 / 68.26** | **65.62 / 63.04** | **59.37 / 58.46** |
| Transcript Only | | 59.50 / 60.47 | 49.77 / 51.43 | 41.32 / 41.58 |
| SpeechCueLLM (Wu et al., 2025) | LLaMA-3-8B-Instruct | 53.41 / 54.41 | 47.10 / 47.77 | 38.46 / 37.69 |
| **VowelPrompt (Ours)** | | **59.14 / 59.67** | **52.29 / 53.49** | **42.50 / 42.64** |

### A.6 ABLATION STUDY ON THE PROSODY-DRIVEN EMOTION PREDICTION IN VOWELPROMPT

To demonstrate that VowelPrompt relies on vowel-level prosodic descriptors instead of spurious lexical or formatting heuristics inherited from the oracle reasoning traces, we conducted a series of ablation studies on transcript shuffle control, prosody permutation control, matched-marginal placebo, and cross-swap. The ablation study is performed on MELD using LLaMA-3-8B-Instruct trained with GRPO. In the study on transcript shuffle control, we randomly permute the word order while

keeping the vowel-level prosodic descriptors intact. It is observed in Table 13 that the performance of VowelPrompt only marginally decreases under this perturbation, indicating that lexical ordering or content identity is not the dominant predictive signal, and the prediction of VowelPrompt heavily relies on the vowel-level prosodic information coupled with the lexical information of the vowels. In the study on prosody permutation control, we permute the vowel-prosody descriptors across utterances within each training mini-batch while leaving transcripts unchanged. It is observed in Table 13 that the prosody permutation leads to a significant performance degradation, which demonstrates that VowelPrompt significantly depends on the alignment between vowel-level prosodic cues and the corresponding utterances, instead of relying on the transcript alone. We further perform a matched-marginal placebo experiment, where prosody tokens are replaced with random draws from their empirical per-vowel distributions. This preserves the marginal statistics, token frequencies, and style patterns but destroys semantic grounding. It is observed in Table 13 that the performance of the ablation model decreases significantly, which demonstrates that VowelPrompt does not rely on superficial token regularities and instead requires aligned prosodic descriptors to make accurate predictions.

Table 13: Ablation study on the prosody-driven emotion prediction in VowelPrompt. The study is performed on MELD using LLaMA-3-8B-Instruct trained with GRPO.

| Methods | Weighted F1 (%) |
|---|---|
| VowelPrompt (Prosody Permutation) | 41.72 |
| VowelPrompt (Matched-Marginal Placebo) | 44.10 |
| VowelPrompt (Transcript Shuffle) | 67.00 |
| **VowelPrompt** | **68.90** |

Finally, we perform a cross-swap counterfactual consistency experiment, where we preserve the transcript but attach prosodic descriptors extracted from utterances belonging to a different emotion category. The study is performed on the happy and the sad emotions in IEMOCAP. It is observed that the predicted emotion systematically follows the swapped prosodic profile rather than the lexical content. As shown in Table 14, when happy utterances are paired with sad prosody, the proportion of predictions labeled as sad increases from $18.7\%$ to $45.8\%$, while retaining the original transcript. Conversely, when sad utterances are paired with happy prosody, the proportion of happy predictions increases from $27.5\%$ to $51.0\%$. The above results demonstrate that VowelPrompt does not merely memorize lexical patterns but actively attributes emotional prediction to vowel-level prosodic cues, which provides direct causal evidence that the prosodic descriptors, rather than text alone, drive the model's decision-making.

Table 14: Counterfactual cross-swap analysis on the happy and the sad emotions.

| Ground-Truth Emotion | Prosody Source | Predicted Happy (%) | Predicted Sad (%) |
|---|---|---|---|
| Happy | Happy | 81.3 | 18.7 |
| Happy | Sad | 54.2 | 45.8 |
| Sad | Sad | 27.5 | 72.5 |
| Sad | Happy | 51.0 | 49.0 |

## A.7    STUDY ON THE IMPACT OF THE TOKENIZATION OF THE LABEL VERBALIZER

To study the impact of the tokenization behavior of the labels, we first perform a study on the tokenization behavior of the IEMOCAP verbalizers, including angry, excited, happy, neutral, and sad, under both the LLaMA-3-8B and Qwen-2-7B tokenizers. In both models, happy, neutral, and sad are each encoded as single-token verbalizers, while angry (['ang','ry']) and excited (['exc', 'ited']) are consistently split into two subword units. To study the impact of the decoding bias arising from such variation, we replaced all emotion labels with synthetic two-letter tokens (happy→ha, sad→sa, angry→an, neutral→ne, excited→ex) that are uniformly represented as single tokens across both tokenizers. We then randomly permuted the emotion verbalizer mapping 10 times, thereby eliminating any lexical or semantic prior that the tokenizer could exploit. It is observed in Table 15 that replacing the original emotion verbalizers with synthetic two-letter tokens leads to only a marginal performance drop on VowelPrompt, which demonstrates that VowelPrompt is marginally impacted

by the decoding bias. Notably, the impact is significantly smaller for VowelPrompt compared to SpeechCueLLM, which demonstrates that VowelPrompt is significantly more robust to label perturbations because the predictions are grounded in detailed vowel-level prosodic cues rather than lexical or tokenization-based priors associated with the verbalizers.

Table 15: Impact of label tokenization and permutation on emotion recognition performance of VowelPrompt.

| Methods | Weighted F1 (%) |
|---|---|
| VowelPrompt | 73.0 |
| SpeechCueLLM | 71.5 |
| VowelPrompt (Two-Letter) | 71.7 |
| SpeechCueLLM (Two-Letter) | 67.8 |
| VowelPrompt (Two-Letter Permutated) | 71.0 |
| SpeechCueLLM (Two-Letter Permutated) | 65.2 |

### A.8 COMPARISON BETWEEN VOWEL-LEVEL AND CONSONANT-LEVEL PROSODIC DESCRIPTORS

To demonstrate the effectiveness of the vowel-level prosodic descriptors, we perform an ablation study replacing vowel-level descriptors with consonant-level descriptors, including segment duration, voice onset time (VOT), frication energy, and nasal intensity. We have also tested the performance of a variant of VowelPrompt, which incorporates both the vowel-level descriptors and the consonant-level descriptors. It is observed in Table 16 that vowel-level descriptors consistently achieve the highest performance across all three corpora, while consonant-level descriptors alone yield significantly worse performance. Incorporating both vowel- and consonant-level cues improves the performance of VowelPrompt on German, which is attributed to the richer stop and fricative contrasts in its phonological system, but does not surpass the vowel-only setting on French or English. These results indicate that consonantal information does not provide universally complementary emotional cues and further substantiate the sufficiency of vowel-level descriptors for the languages evaluated in this study.

Table 16: Comparison between vowel-level with consonant-level prosodic descriptors.

| Methods | CaFE (French) | EmoDB (German) | MELD (English) |
|---|---|---|---|
| VowelPrompt (Vowel-Level Cues) | 51.42 | 69.85 | 64.17 |
| VowelPrompt (Consonant-Level Cues) | 48.73 | 67.80 | 62.95 |
| VowelPrompt (Vowel + Consonant Cues) | 51.04 | 70.21 | 64.08 |

### A.9 COMPARISON WITH NON-LLM ACOUSTIC BASELINE METHODS

To demonstrate the advantages of VowelPrompt over existing non-LLM deep learning models, we have compared VowelPrompt with strong non-LLM speech emotion recognition baselines using state-of-the-art self-supervised speech models, including HuBERT-large and wav2vec-large. In particular, we extract HuBERT-large and wav2vec-large embeddings and train MLP classifiers on top of the embeddings. The results below include both in-domain and cross-domain evaluations on IEMO-CAP and MELD. It is observed in Table 17 that VowelPrompt consistently outperforms HuBERT-large and wav2vec-large across all settings.

### A.10 HUMAN EVALUATION ON THE REASONING TRACES BY VOWELPROMPT

To evaluate the quality of the reasoning traces of VowelPrompt trained with GRPO, we conducted a human evaluation study with four annotators who rated 200 randomly sampled reasoning traces from each of the models trained on IEMOCAP and MELD. Each trace was evaluated on prosodic grounding, causal coherence, and internal consistency using a 1–5 Likert scale. It is observed in Table 18 that VowelPrompt demonstrates significantly higher reasoning faithfulness than SpeechCueLLM across all four annotators. SpeechCueLLM receives an average score of 3.14, while VowelPrompt

Table 17: Comparison of VowelPrompt with strong non-LLM acoustic baselines using self-supervised speech representations calculated from HuBERT-large and wav2vec-large.

| Datasets | HuBERT-large | wav2vec-large | VowelPrompt |
|---|---|---|---|
| In-Domain Evaluations | | | |
| IEMOCAP | 67.6 | 65.6 | 73.4 |
| MELD | 56.8 | 55.1 | 69.6 |
| Cross-Domain Evaluations | | | |
| IEMOCAP → MELD | 45.0 | 43.5 | 60.2 |
| MELD → IEMOCAP | 44.2 | 41.7 | 51.7 |

receives an average score of 3.77, reflecting more accurate grounding in prosodic cues and greater internal coherence. These results confirm that VowelPrompt's reasoning traces are not only more interpretable but also more consistently aligned with the prosodic evidence that drives its final predictions.

Table 18: Human evaluation results across four evaluators.

| Methods | Evaluator 1 | Evaluator 2 | Evaluator 3 | Evaluator 4 | Average |
|---|---|---|---|---|---|
| SpeechCueLLM | 3.12 | 3.55 | 2.82 | 3.08 | 3.14 |
| VowelPrompt (Ours) | 4.05 | 3.96 | 3.42 | 3.65 | 3.77 |

## A.11 STUDY ON THE BALANCE BETWEEN THE IN-DOMAIN PERFORMANCE AND THE CROSS-DOMAIN PERFORMANCE

The Knowledge Distillation (KL) weight in GRPO controls how strongly the policy is regularized toward the supervised SFT model, which effectively limits how far reinforcement learning can deviate from the source-domain distribution. To better understand the trade-off between the in-domain performance and the cross-domain performance, we conducted a sensitivity analysis by varying the KL weight and measuring both in-domain and cross-domain performance. As shown in Table 19, decreasing the KL weight relaxes the constraint on the policy, resulting in slightly improved cross-domain robustness, indicating that the model relies less on dataset-specific lexical patterns and more on domain-invariant prosodic cues. On the other hand, increasing the KL weight leads to higher in-domain performance. Notably, both the in-domain and cross-domain performance of VowelPrompt vary only marginally across different values of the KL weight, which demonstrates that the GRPO-trained model is largely insensitive to the value of the KL weight. In addition, in the cross-domain setting, the GRPO data is from the source domain alone.

Table 19: Effect of KL weight on VowelPrompt performance under GRPO for both in-domain and cross-domain evaluation.

| KL Weight | IEMOCAP | MELD | IEMOCAP→MELD | MELD→IEMOCAP |
|---|---|---|---|---|
| 0.1 | 71.9 | 68.1 | 60.5 | 51.3 |
| 0.25 | 73.4 | 69.6 | 60.2 | 51.7 |
| 0.5 | 73.4 | 69.9 | 58.9 | 49.6 |
| 1.0 | 73.6 | 70.0 | 58.4 | 49.2 |

## A.12 COMPARISON WITH CLASSIFIERS TRAINED DIRECTLY ON THE VOWEL-LEVEL PROSODIC FEATURES

To demonstrate the necessity of an LLM-based architecture, we have conducted an ablation study comparing VowelPrompt against classifiers trained directly on the vowel-level prosodic features. In particular, the classifiers are a multilayer perceptron (MLP), a random forest (RF), and a transformer. In the transformer baseline, each phoneme is treated as a token, and its corresponding prosodic features are treated as the features of the token. It is observed in Table 20 that VowelPrompt significantly outperforms all baseline classifiers, which demonstrates that access to the same attributes alone is

insufficient. Vanilla classifiers fail to capture the contextual and linguistic dependencies that govern how vowel-level prosody conveys affect. In contrast, VowelPrompt leverages the LLM's pretrained linguistic priors to integrate prosodic cues with lexical semantics, discourse context, and phonotactic patterns.

Table 20: Comparison with traditional classifiers trained directly on the vowel-level prosodic features.

| Datasets | XGBoost | MLP | Transformer | VowelPrompt |
|----------|---------|------|-------------|-------------|
| IEMOCAP  | 40.2    | 39.6 | 48.5        | 73.4        |
| MELD     | 45.1    | 44.5 | 51.2        | 69.6        |

### A.13 STUDY ON THE IMPACT OF THE INCORRECT VOWEL ALIGNMENT

To study the impact of incorrect vowel alignment, we have performed an ablation study that perturbed 5%, 10%, and 15% of the boundaries of the vowels in the alignment results. The study is performed on MELD using LLaMA-3-8B-Instruct. In particular, for each selected vowel segment, we randomly shifted its start or end times by 50% of its original duration. It is observed in Table 21 that the performance of VowelPrompt is robust to the perturbation of the boundaries of the vowels in the alignment results and consistently achieves significantly better performance than SpeechCueLLM. For example, even with 15% of the vowel boundaries perturbed, VowelPrompt still achieves a Weighted F1 of 69.11%, which outperforms SpeechCueLLM by 2.04%.

Table 21: Robustness of VowelPrompt under phenom alignment perturbations.

| Method | Perturbation Ratio | Weighted F1 (%) |
|--------|--------------------|-----------------|
| SpeechCueLLM | 0   | 67.07 |
| VowelPrompt  | 0   | 69.61 |
| VowelPrompt  | 5%  | 69.50 |
| VowelPrompt  | 10% | 69.23 |
| VowelPrompt  | 15% | 69.11 |

### A.14 STUDY ON THE IMPACT OF SPEECH RATE ON THE PERFORMANCE OF VOWELPROMPT

To study the impact of speech rate on the performance of VowelPrompt, we conducted an ablation study on the MELD dataset by categorizing testing utterances according to their phone-per-second (PPS) rate. In particular, we segmented the test set into three categories based on PPS statistics computed from Montreal Forced Aligner alignments, including slow (PPS $\leq 6.0$), normal ($6.0 <$ PPS $\leq 8.5$), and fast (PPS $> 8.5$). It is observed in Table 22 that although the performance of VowelPrompt and the baseline method SpeechCueLLM degrade as speech rate increases, VowelPrompt consistently outperforms SpeechCueLLM across all PPS ranges.

Table 22: Impact of speech rate (phones-per-second, PPS) on the performance of VowelPrompt.

| Method | PPS $\leq 6.0$ | $6.0 <$ PPS $\leq 8.5$ | PPS $> 8.5$ | Overall |
|--------|----------------|------------------------|-------------|---------|
| SpeechCueLLM | 68.21 | 67.44 | 64.19 | 67.07 |
| VowelPrompt  | 71.08 | 69.61 | 67.25 | 69.61 |

### A.15 ABLATION STUDY ON INCORPORATING THE REASONING IN SFT AND SFT COMBINED WITH GRPO

In this section, we perform an ablation study by enabling and disabling reasoning/thinking in both the SFT and the SFT & GRPO settings. The study is performed on MELD using LLaMA-3-8B-Instruct. It is observed in Table 23 that our VowelPrompt consistently outperforms the baseline methods under different settings.

Table 23: Comparison of different training strategies with and without reasoning.

| Method | SFT (w/o Reasoning) | SFT (with Reasoning) | SFT & GRPO (w/o Reasoning) | SFT & GRPO (with Reasoning) |
|---|---|---|---|---|
| InstructERC | 67.25 | 67.02 | 67.51 | 66.96 |
| SALMONN | 67.25 | 67.25 | 67.43 | 66.85 |
| SpeechCueLLM | 67.07 | 67.15 | 67.29 | 67.10 |
| VowelPrompt (Ours) | 69.61 | 69.82 | 69.88 | 68.98 |

## B PROMPT TEMPLATES

We present representative prompt templates used in our experiments across zero-shot, few-shot, and fine-tuning regimes. Each prompt includes three main components: the conversational context, the target utterance, and the prosodic descriptors (either sentence-level or vowel-level). Descriptors are expressed in natural language and inserted into prompts using a consistent format to guide emotion reasoning.

**Supervised Fine-Tuning Prompt (with Reasoning):**

> Now you are an expert in sentiment and emotional analysis.
> The following conversation noted between '### ###' involves several speakers.
> ### Speaker_0:...
> ...
> Speaker_1:<target_speech> ###
> Vowel-level Speech Descriptions of Speaker_1:<target_speech>:
> <vowel_descriptions>
> Please select the emotional label of Speaker_1:<target_speech> based on the context and the vowel-level acoustic features.
> Output the thinking process in <think> </think> and emotion label prediction in <answer> </answer> tags.

**Supervised Fine-Tuning Prompt (without Reasoning):**

> Now you are an expert in sentiment and emotional analysis.
> The following conversation noted between '### ###' involves several speakers.
> ### Speaker_0:...
> ...
> Speaker_1:<target_speech> ###
> Vowel-level Speech Descriptions of Speaker_1:<target_speech>:
> <vowel_descriptions>
> Please select the emotional label of Speaker_1:<target_speech> based on the context and the vowel-level acoustic features.
> Output the emotion label prediction in <answer> </answer> tags.

**Zero-Shot Prompt (Transcript Only):**

> Now you are an expert in sentiment and emotional analysis.
> The following conversation noted between '### ###' involves several speakers.
> ### Speaker_0:...
> ...
> Speaker_1:<target_speech> ###
> Please select the emotional label of Speaker_1:<target_speech> based on the context.
> Please output ONLY ONE label from <available_emotion_labels> as the first word, and then explain your choice.

**Zero-Shot Prompt (Transcript + Vowel-Level Prosody):**

Now you are an expert in sentiment and emotional analysis.
The following conversation noted between '### ###' involves several speakers.
### `Speaker_0:...`
...
`Speaker_1:<target_speech>` ###
Vowel-level Speech Descriptions of `Speaker_1:<target_speech>`:
`<vowel_descriptions>`
Please select the emotional label of `Speaker_1:<target_speech>` based on the context and the vowel-level acoustic features.
Please output ONLY ONE label from `<available_emotion_labels>` as the first word, and then explain your choice.

**Few-Shot Prompt (3 Examples + Target Query)**:

Now you are an expert in sentiment and emotional analysis.

`<Example_1>`

`<Example_2>`

`<Example_3>`

The following conversation noted between '### ###' involves several speakers.
### `Speaker_0:...`
...
`Speaker_1:<target_speech>` ###
Vowel-level Speech Descriptions of `Speaker_1:<target_speech>`:
`<vowel_descriptions>`
Please select the emotional label of `Speaker_1:<target_speech>` based on the context and the vowel-level acoustic features.
Please output ONLY ONE label from `<available_emotion_labels>` as the first word, and then explain your choice.

