# OpenReview forum: "VowelPrompt: Hearing Speech Emotions from Text via Vowel-level Prosodic Augmentation"
_ICLR.cc/2026/Conference — ICLR 2026 Poster_

### Official Review · Reviewer_b43y · 2025-10-27

**Soundness:** 2
**Presentation:** 3
**Contribution:** 3
**Rating:** 4
**Confidence:** 5

**Summary:**

This paper described a system that augments vowel-level prosody descriptors as additional information to the LLMs. It shows that with the vowel-level speech information, the LLMs performs better on emotion recognition in both zero-shot and supervised fine-tuning setup.

**Strengths:**

The paper is well written with good clarity, and easy to understand. It could contribute to the SER field as a baseline of fine-tuning LLMs with natural language descriptors of speech. I am personally interested in what granularity of speech information is needed for SER. This paper seems to suggests that going toward vowel-level will provide some improvement.

**Weaknesses:**

On the other hand, the idea of appending prosody descriptions itself is not particularly novel, e.g., SpeechCueLLM, which is also cited as a baseline by the authors. The novelty of the work lies in expanding it to the vowel-level.

There is one flaw that undermines the validity of the experiments that the work needs to revise before publication.
In Table 4, it is unclear to me how the other models are trained and tested. I assume only your model is fine-tuned with the thinking trajectory but not the other baselines, as I don't find how you generate the thinking trajectories for the baselines. In this case, I don't know if the other models will still output thinking trajectory after fine-tuning. And the performance gain you get, may simply be a difference of inference time scaling. And it can be possible that if you make the baseline models think/reason, they can also improve their performance to be potentially on par with your model.

The authors should add experiments either comparing the proposed model v.s. the baselines in either (both thinking enabled, or both thinking disabled). I am willing to raise the score if this issue is resolved.

Additionally, I don't see adding GRPO helps in Table 4, the authors should explain this.

Typo: in the abstract: neglect "or"

**Questions:**

See the above weaknesses.

---

> ### Author Response · Authors · 2025-12-03
> **Response to the Official Review by Reviewer b43y (Part 1)**
>
> We appreciate the review and the suggestions in this review. The raised issues are addressed below.
>
> **Responses to the Weaknesses**
>
> **1. ”On the other hand, the idea of appending prosody descriptions itself is not particularly novel, e.g., SpeechCueLLM, which is also cited as a baseline by the authors. The novelty of the work lies in expanding it to the vowel-level.”**
>
> We clarify that the contribution of VowelPrompt is not only the addition of prosodic descriptors to the input, as explored in SpeechCueLLM, but the introduction of a reasoning-centric framework that enables the model to perform structured inference over fine-grained, vowel-level prosodic cues. Although SpeechCueLLM [1] provides coarse, sentence-level prosodic annotations that function as auxiliary features, VowelPrompt transforms prosodic information into phoneme-aligned fine-grained descriptors that are explicitly invoked within the model’s internal reasoning process (“<think>…</think>”).
> As shown in our response to Weakness 2, VowelPrompt consistently outperforms SpeechCueLLM with and without reasoning enabled, which demonstrates that the gains are not because of the presence of prosodic features, but due to the model’s ability to causally integrate and reason over these fine-grained cues.
> Moreover, the benefits of such fine-grained reasoning capability are even more significant under the cross-domain transfer setting, as shown in Table 5 of our paper, where VowelPrompt achieves significant improvements over SpeechCueLLM. For example, VowelPrompt outperforms SpeechCueLLM by 6.96% in weighted F1 in the MELD→IEMOCAP transfer setting, which shows that the model’s reasoning over vowel-level prosodic structure provides domain-invariant informative signals that generalize beyond lexical or dataset-specific patterns.
>
>
> **2. ”...There is one flaw that undermines the validity of the experiments that the work needs to revise before publication. In Table 4, it is unclear to me how the other models are trained and tested.… The authors should add experiments either comparing the proposed model v.s. the baselines in either (both thinking enabled, or both thinking disabled). I am willing to raise the score if this issue is resolved.”**
>
> For the SFT setting in Table 4, the reasoning is not incorporated into the training and inference processes. We train both the baseline methods and our VowelPrompt following the settings in SpeechCueLLM [1]. For the SFT &GRPO setting in Table 4, reasoning is incorporated for both the baseline methods and our VowelPrompt during both training and inference processes. All methods are first warmed up using gold reasoning traces generated by GPT-4o on 20% of the training data. All methods are trained with GRPO using the same reward function in Equation (1) of our paper, which jointly optimizes prediction accuracy and adherence to the prescribed output format (<think>...</think> and <answer>...</answer>). During inference, all models produce and are evaluated with the same reasoning-enabled output structure.
>
> Furthermore, we have performed an ablation study by enabling and disabling reasoning/thinking in both the SFT and the SFT&GRPO settings. The study is performed on MELD using LLaMA-3-8B-Instruct. It is observed in the table below that our VowelPrompt consistently outperforms the baseline methods under different settings.
>
> | Method             | SFT (w/o Reasoning) | SFT (with Reasoning) | SFT & GRPO (w/o reasoning) | SFT & GRPO (with Reasoning) |
> | --- | :----: | :----: | :----: | :----: |
> | InstructERC        |         67.25     |     67.02  |     67.51 |            66.96      |
> | SALMONN   |         67.25         |        67.25        |   67.43     |   66.85    |
> | SpeechCueLLM   |     67.07     |        67.15      |  67.29       |    67.10        |
> | VowelPrompt (Ours) |         **69.61**         |     **69.82**     |   **69.88**     | **68.98**            |
>
> The ablation study results above are also added to Table 23 in Section A.15 of the revised paper.

---

> ### Author Response · Authors · 2025-12-03
> **Response to the Official Review by Reviewer b43y (Part 2)**
>
> **3. ”Additionally, I don't see adding GRPO helps in Table 4, the authors should explain this.”**
>
> GRPO is not intended to universally improve performance in fully supervised settings. Instead, its primary contribution lies in refining model behavior when explicit reasoning is present by enforcing structural correctness and reward-aligned inference. GRPO provides no additional supervisory signal beyond the ground-truth label, which is already used in the SFT setting. As a result, accuracy improvements in this regime remain limited, which is fully consistent with recent analyses demonstrating that SFT largely determines in-distribution performance, while RLVR with GRPO contributes to robustness beyond the training distribution [2].
> When reasoning is enabled, GRPO introduces a verifiable reward that incentivizes coherent reasoning chains and adherence to the prescribed output format. This reinforcement signal induces more stable and systematic decision-making, which is significantly effective under a distributional shift. As shown in Table 5 of our paper, GRPO leads to substantial performance improvements in the cross-domain experiments compared to SFT. Moreover, VowelPrompt trained with GRPO is significantly more effective than the baseline methods trained with GRPO. For example, VowelPrompt outperforms SpeechCueLLM by 6.96% in the MELD → IEMOCAP transfer setting. Such improvements are attributed to the fine-grained, vowel-level prosodic descriptors introduced by VowelPrompt, which provide richer and more localized affective cues than coarse sentence-level prosodic summaries. As a result, GRPO can exploit the fine-grained informative prosodic structures more effectively, thereby enhancing reasoning quality and cross-domain generalization.
>
> **4 ”Typo: in the abstract: neglect "or".”**
>
> We have revised the typo in the revised paper.
>
> **References**
>
> [1] Wu, Zehui, et al. "Beyond silent letters: Amplifying llms in emotion recognition with vocal nuances." Findings of the Association for Computational Linguistics: NAACL 2025. 2025.
>
> [2] Kirk, Robert, et al. "Understanding the Effects of RLHF on LLM Generalisation and Diversity." ICLR 2024.

---

### Official Review · Reviewer_W37J · 2025-11-01

**Soundness:** 2
**Presentation:** 3
**Contribution:** 3
**Rating:** 4
**Confidence:** 5

**Summary:**

The paper presents an approach to introduce an explainable and interpretable approach to perform emotion recognition from speech using  LLMs. The work introduces pitch, energy, and duration based descriptors from time-aligned vowel segments and converts these into natural language descriptors to fine tune existing LLMs. The authors hypothesize that the proposed descriptors would enable an LLM to reason using both semantic information in lexical representations as well as prosodic variations in acoustic information.

Evaluation on multiple datasets demonstrate that the proposed approach helped to improve performance compared to prior art.

The motivation of the paper is well outlined, clearly citing relevant prior art and outlining the key contributions made in this paper.
The proposed approach introduces prosodic descriptors based on their proposed approach, that are generated from standard benchmark datasets, it is not clear whether the authors intend to share the information with the community as that would help in both replication of the reported results and foster future research directions.

**Strengths:**

The paper introduces pitch, energy, and duration based descriptors from time-aligned vowel segments to generate emotion-salient prompts to improve emotion recognition performance using LLMs.

The process of generating the descriptors is well described and results demonstrate the promise of the proposed work. Evaluation from multiple datasets demonstrate the generalization of the findings.

**Weaknesses:**

The proposed approach introduces prosodic descriptors based on their proposed approach, that are generated from standard benchmark datasets, it is not clear whether the authors intend to share the information with the community as that would help in both replication of the reported results and foster future research directions. It is not clear whether the setup used to generate the descriptors, or the descriptors obtained from the five datasets (used in the paper) will be publicly shared.

There are some open questions regarding the introduction of the descriptors that need to be addressed: coarticulation and lenition in spontaneous speech can alter vowel durations where vowels can be deleted or influenced by neighboring phonemes. It is not clear how the proposed approach would address such situations, or such instances were ignored in the current work? It will also be interesting to explore how speech rate impacts the efficacy of using the proposed descriptors?

**Questions:**

(1) By vowel level, does it mean at the individual distinct vowel level or by vowel-groups, such as front and back vowels?

(2) For the cross-domain evaluations, the datasets had different number of emotion categories, how were the difference in categories accounted during the evaluation?

(3) When reporting the emotion recognition performance across the different datasets, were all emotion categories specified in those data sets (Table 2) used to obtain the evaluation metric?

(4) The descriptors presented in this work rely on the forced alignment information, in spontaneous speech due to coarticulation vowel boundaries may be uncertain, any thoughts on how such conditions may impact the proposed descriptors?

---

> ### Author Response · Authors · 2025-12-03
> **Response to the Official Review by Reviewer W37J (Part 1)**
>
> We appreciate the review and the suggestions in this review. The raised issues are addressed below.
>
> **Responses to the Weaknesses**
>
> **1. ”...It is not clear whether the setup used to generate the descriptors … will be publicly shared.”**
>
> As described in Sections 3.1 and 3.3 of our paper, all prosodic descriptors in our framework are generated using publicly available tools and standardized phonetic resources. The phoneme-level temporal alignments are obtained using the Montreal Forced Aligner (MFA) [1]. The vowel identities across languages are aligned using the International Phonetic Alphabet (IPA) [2]. The calculation of the acoustic descriptors, including pitch, intensity, and duration, is performed using the widely adopted tool, Praat.
>
> **2. ”...coarticulation and lenition in spontaneous speech can alter vowel durations, where vowels can be deleted or influenced by neighboring phonemes. … It will also be interesting to explore how speech rate impacts the efficacy of using the proposed descriptors?”**
>
> As described in Section 3.1, vowel durations and prosodic descriptors are extracted after phoneme-level forced alignment using the Montreal Forced Aligner (MFA), which provides context-dependent alignment boundaries. This alignment inherently accounts for phonetic reduction phenomena such as vowel centralization, partial lenition, and durational shortening induced by adjacent consonants. Moreover, all descriptors are subsequently normalized at both the speaker level and the vowel-class level, which reduces sensitivity to absolute duration shifts introduced by coarticulation and phonological weakening.
> To mitigate the negative impacts of the deleted or heavily reduced vowels, our pipeline follows the standard practice in computational phonetics to remove the vowels that last less than 50 ms.
>
> To study the impact of speech rate on the performance of VowelPrompt, we conducted an ablation study on the MELD dataset by categorizing testing utterances according to their phone-per-second (PPS) rate. In particular, we segmented the test set into three categories based on PPS statistics computed from Montreal Forced Aligner alignments, including slow (PPS $\leq$ 6.0), normal (6.0 $<$ PPS $\leq$ 8.5), and fast (PPS $>$ 8.5). It is observed in the table that although the performance of VowelPrompt and the baseline method SpeechCueLLM degrade as speech rate increases, VowelPrompt consistently outperforms SpeechCueLLM across all PPS ranges.
>
> | Method | PPS≤6.0| 6.0 < PPS ≤ 8.5 | PPS > 8.5 | Overall |
> | --- | --- | --- | --- | --- |
> | SpeechCueLLM | 68.21               | 67.44                    | 64.19               | 67.07   |
> | VowelPrompt  | 71.08               | 69.61                    | 67.25               | 69.61   |
>
> The ablation study results above are also added to Table 22 in Section A.14 of the revised paper.
>
>
> **Responses to the Questions**
>
> **1. ”By vowel level, does it mean at the individual distinct vowel level or by vowel-groups, such as front and back vowels?”**
>
> As described in Section 3.1 of our paper, vowel-level refers to individual vowel segments aligned at the phoneme level, rather than coarse-grained vowel groupings such as front/back or high/low categories. Each vowel instance in the utterance is extracted based on forced-alignment boundaries, and prosodic descriptors (duration, pitch trajectory, and intensity profile) are computed for that specific segment.
>
> **2. ”For the cross-domain evaluations, the datasets had different number of emotion categories, how were the difference in categories accounted during the evaluation?”**
>
> In the cross-domain evaluation, we construct a unified label space by mapping semantically equivalent categories across the two corpora, following existing work [6]. Moreover, having a different number of emotion categories does not pose an issue in our setup, because the evaluation prompt explicitly enumerates the emotion categories available in the current domain. As a result, the model is only allowed to predict from the valid label set specified in the prompt, regardless of how many categories exist in the source dataset.
>
> **3. ”When reporting the emotion recognition performance across the different datasets, were all emotion categories specified in those data sets (Table 2) used to obtain the evaluation metric?”**
>
> Yes, we use all emotion categories specified in each dataset, as listed in Table 2, to compute the evaluation metrics.

---

> > ### Author Response · Authors · 2025-12-03
> > **Response to the Official Review by Reviewer W37J (Part 2)**
> >
> > **4. ”The descriptors presented in this work rely on the forced alignment information, in spontaneous speech due to coarticulation vowel boundaries may be uncertain, any thoughts on how such conditions may impact the proposed descriptors?”**
> >
> > Although coarticulation can introduce minor temporal variability at segment boundaries, vowel nuclei, which is the primary carriers of prosodic information, remain acoustically stable regions characterized by sustained voicing and reliable formant structure [7]. Because our descriptors aggregate pitch, intensity, and duration over these nuclei rather than depending on exact phoneme onset or offset locations, they are intrinsically robust to such boundary imprecision.
> > Moreover, to study the impact of incorrect vowel boundaries, we have performed an ablation study that perturbed 5%, 10%, and 15% of the boundaries of the vowels in the alignment results. The study is performed on MELD using LLaMA-3-8B-Instruct. In particular, for each selected vowel segment, we randomly shifted its start or end times by 50% of its original duration. It is observed in the table below that the performance of VowelPrompt is robust to the perturbation of the boundaries of the vowels in the alignment results and consistently achieves significantly better performance than SpeechCueLLM. For example, even with 15% of the vowel boundaries perturbed, VowelPrompt still achieves a Weighted F1 of 69.11%, which outperforms SpeechCueLLM by 2.04 %.
> > | Method | Perturbation Ratio | Weighted F1 |
> > | --- | --- | --- |
> > | SpeechCueLLM |0 | 67.07 %  |
> > | VowelPrompt  |0 | 69.61 %  |
> > | VowelPrompt  |5% |  69.50 %|
> > | VowelPrompt  |10% |  69.23 % |
> > | VowelPrompt  |15% |  69.11 % |
> >
> >
> >
> > **References**
> >
> > [1] McAuliffe, Michael, et al. "Montreal forced aligner: Trainable text-speech alignment using kaldi." Interspeech. Vol. 2017. 2017.
> >
> > [2] International Phonetic Association. Handbook of the International Phonetic Association: A guide to the use of the International Phonetic Alphabet. Cambridge University Press, 1999.
> >
> > [3] Paul Boersma and David Weenink. Praat, a system for doing phonetics by computer. Glot International, 5(9/10):341–345, 2001.
> >
> > [4] Kendall, Tyler, and Valerie Fridland. Sociophonetics. Cambridge University Press, 2021.
> >
> > [5] Stevenson, Thomas, Catherine Rees, and Simon Hodder. "The Acoustic Profiles of Emotion: Analyzing the Spoken Voice in Theater Performance." Voice and Speech Review (2025): 1-16.
> >
> > [6] Lei, Shanglin, et al. "Instructerc: Reforming emotion recognition in conversation with multi-task retrieval-augmented large language models." arXiv preprint arXiv:2309.11911 (2023).
> >
> > [7] Genette, Jérémy, et al. "Determining spectral stability in vowels: A comparison and assessment of different metrics." Speech Communication 154 (2023): 102984.

---

### Official Review · Reviewer_RfwA · 2025-11-01

**Soundness:** 2
**Presentation:** 3
**Contribution:** 3
**Rating:** 4
**Confidence:** 4

**Summary:**

The paper targets a key limitation in speech emotion recognition (SER) with large language models (LLMs): difficulty leveraging fine-grained prosodic cues and lack of interpretability. The authors propose VowelPrompt, a framework that annotates and conditions LLMs on multi-dimensional, phoneme-level prosodic attributes with a focus on vowels. Experiments across five datasets (including IEMOCAP, MELD, CaFE) show consistent gains over text-only baselines and sentence-level prosody augmentation methods (e.g., SpeechCueLLM) in zero-shot, few-shot, supervised fine-tuning, cross-domain , and multilingual settings, while providing interpretable emotion reasoning.

**Strengths:**

1. Fine-grained modeling unit: The vowel-centric, phoneme-level prosodic prompting is a clear step beyond sentence-level prosody features, improving both recognition performance and interpretability. The focus on vowels is linguistically motivated and empirically supported.
2. Two-stage training design: The two-stage framework is logically structured, and the RL-based targeted optimization appears to contribute to robust cross-domain generalization.
3. Breadth of evaluation: The empirical study is comprehensive, covering zero-shot, few-shot, supervised fine-tuning, cross-domain transfer (IEMOCAP↔MELD), and multilingual scenarios (EN/FR/DE/mixed). The baselines include strong text-only and prosody-augmented LLM variants (e.g., SpeechCueLLM), making the comparisons meaningful.

**Weaknesses:**

1. The paper emphasizes the primacy of vowels for prosody, while citing evidence that consonants can convey complementary emotional cues (e.g., Bitouk et al., 2010). However, the method entirely excludes consonant segments. This design choice risks discarding potentially informative signals (e.g., frication intensity, voicing onsets, burst characteristics) that may be emotion-sensitive in certain languages and speaking styles. A controlled analysis is needed to justify the exclusion.
2. The baseline suite lacks strong non-LLM SER systems, despite a long line of work using acoustic-prosodic features and modern architectures (e.g., CNN/TDNN/Conformer or HuBERT/w2v2 features with phoneme-aligned prosody). Without such baselines, it is difficult to disentangle the benefit of VowelPrompt from the general advantage conferred by LLM priors and instruction tuning.
3. The framework assumes fine-grained (phoneme/vowel-level) labels and intermediate “reasoning” steps align with the final emotion decision. In practice, manual annotations and model-generated rationales may not be fully consistent. This raises concern that the LLM could optimize for correct final labels while producing intermediate explanations that are partially contradictory or post hoc. The paper should explicitly evaluate the faithfulness and consistency of the reasoning traces.
4. Table 4 suggests GRPO leads to consistent performance drops (negative optimization) under certain settings, while Table 5 shows clear cross-domain improvements. The manuscript needs a unified explanation reconciling these outcomes. For instance, is GRPO trading in-domain accuracy for distributional robustness, regularizing prosody usage, or mitigating spurious correlations? What hyperparameters or curricula drive this trade-off?
5. At inference time, the system still relies on MFA and explicitly labeled phoneme attributes before passing prompts to the LLM. It is therefore unclear why an LLM is preferable to a purpose-built classifier over the same discrete features (e.g., a gradient-boosted tree, MLP, or Transformer). If the LLM mainly converts discrete features to natural-language prompts, the incremental contribution appears limited. The paper should quantify the benefit of the LLM beyond a feature-to-label classifier and clarify where language priors or compositional reasoning matter.
5. The framework depends on accurate phoneme segmentation and prosodic feature extraction. It is unclear how errors in forced alignment or noisy conditions propagate to final performance. Robustness analyses (e.g., perturbing boundaries, adding noise, cross-accent conditions) are limited.

**Questions:**

1. Vowel vs. consonant contributions: Can you provide ablations comparing vowel-only, consonant-only, and all-phoneme prompting across languages and conditions? Are there language-specific effects where consonant cues (e.g., voicing contrasts, place/manner-dependent energy) help more?
2. Non-LLM baselines: How does VowelPrompt compare against a strong acoustic SER pipeline using phoneme-aligned prosody with CNN/Conformer or self-supervised speech features (HuBERT/w2v2) plus a classifier? Please include both in-domain and cross-domain comparisons.
3. Reasoning faithfulness: Do you measure the consistency between intermediate prosodic rationales and final predictions? For example, human judgments of faithfulness, agreement with attributions  or causal tests (masking/removing cited cues).
4. GRPO dynamics:  Can you explain or give learning curves and sensitivity analyses demonstrating when GRPO helps harm in-domain performance but helps cross-domain transfer? Also, if I understand correctly, the GRPO data is also from the source domain, right?
5. Value of LLMs: If the same phoneme-level attributes are available, what performance and interpretability gains remain when replacing the LLM with a standard classifier? Conversely, can the LLM operate without explicit phoneme labels (i.e., text + raw audio features) and still retain its advantages?
6. How robust is VowelPrompt to alignment errors and background noise? What is the performance degradation under realistic ASR/aligner noise or accented speech?

---

> ### Author Response · Authors · 2025-12-03
> **Response to the Official Review by Reviewer RfwA (Part 1)**
>
> We appreciate the review and the suggestions in this review. The raised issues are addressed below.
>
> **Responses to the Weaknesses and Questions**
>
> **Weakness 1. ”...However, the method entirely excludes consonant segments … A controlled analysis is needed to justify the exclusion.”**
> **Question 1. ”Vowel vs. consonant contributions: Can you provide ablations comparing vowel-only, consonant-only, and all-phoneme prompting across languages and conditions? …”**
>
> Our design centers on vowels because vowel nuclei constitute the most reliable and acoustically stable carriers of prosodic variation, including pitch, intensity, and duration, across spontaneous and multilingual speech settings [1]. Consonantal cues such as frication noise, burst energy, and voicing onsets are typically considered as brief, lower-intensity events with rapidly changing spectral structure, in contrast to the longer and more stationary vowel nuclei. Prior phonetic work [2] shows that consonants are generally briefer and less intense than vowels and often require multiple dynamic spectral measures to characterize place and manner reliably, particularly for fricatives, which exhibit highly variable noise spectra.
> In addition, we performed an ablation study replacing vowel-level descriptors with consonant-level descriptors, including segment duration, voice onset time (VOT), frication energy, and nasal intensity. We have also tested the performance of a variant of VowelPrompt, which incorporates both the vowel-level descriptors and the consonant-level descriptors. It is observed in the table below that vowel-level descriptors consistently achieve the highest performance across all three corpora, while consonant-level descriptors alone yield significantly worse performance. Incorporating both vowel- and consonant-level cues improves the performance of VowelPrompt on German, which is attributed to the richer stop and fricative contrasts in its phonological system, but does not surpass the vowel-only setting on French or English. These results indicate that consonantal information does not provide universally complementary emotional cues and further substantiate the sufficiency of vowel-level descriptors for the languages evaluated in this study.
>
> | Methods      | CaFE (French) | EmoDB (German) | MELD (English) |
> | --- | ---- | --- | --- |
> | VowelPrompt (Vowel-Level Cues)       | 51.42         | 69.85          | 64.17          |
> | VowelPrompt (Consonant-Level Cues)                 | 48.73         | 67.8
> | VowelPrompt (Vowel-Level and Consonant-Level Cues) | 51.04     | 70.21      | 64.08     |
>
> The ablation study results above are also added to Table 16 in Section A.8 of the revised paper.
>
> **Weakness 2. ”The baseline suite lacks strong non-LLM SER systems, despite a long line of work using acoustic-prosodic features and modern architectures (e.g., CNN/TDNN/Conformer or HuBERT/w2v2 features with phoneme-aligned prosody) ….”**
> **Question 2. ”Non-LLM baselines: How does VowelPrompt compare against … self-supervised speech features (HuBERT/w2v2) plus a classifier? ...”**
>
> To demonstrate the advantages of VowelPrompt over existing non-LLM deep learning models, we have compared VowelPrompt with strong non-LLM speech emotion recognition baselines using state-of-the-art self-supervised speech models, including HuBERT-large [3] and wav2vec2-large [4]. In particular, we extract HuBERT-large and wav2vec2-large embeddings and train MLP classifiers on top of the embeddings. The results below include both in-domain and cross-domain evaluations on IEMOCAP and MELD. It is observed in the tables below that VowelPrompt consistently outperforms HuBERT-large and wav2vec2-large across all settings.
>
> | Datasets | HuBERT-large | wav2vec2-large | VowelPrompt |
> | --- | ---- | --- | --- |
> | IEMOCAP        | 67.6  | 65.6   | 73.4      |
> | MELD              |  56.8     |   55.1     | 69.6      |
>
>
> | Datasets     | HuBERT-large | wav2vec2-large | VowelPrompt |
> | ------------ | ------------ | ------------- | ----------- |
> | IEMOCAP→MELD |     45.0         |       43.5        | 60.2        |
> | MELD→IEMOCAP |     44.2        |         41.7      | 51.7        |
>
> The ablation study results above are also added to Table 17 in Section A.9 of the revised paper.

---

> ### Author Response · Authors · 2025-12-03
> **Response to the Official Review by Reviewer RfwA (Part 2)**
>
> **Weakness 3. ”... The paper should explicitly evaluate the faithfulness and consistency of the reasoning traces.”**
> **Question 3. ”Reasoning faithfulness: Do you measure the consistency between intermediate prosodic rationales and final predictions? …”**
>
> To evaluate the quality of the reasoning traces of VowelPrompt trained with GRPO, we conducted a human evaluation study with four annotators who rated 200 randomly sampled reasoning traces from each of the models trained on IEMOCAP and MELD. Each trace was evaluated on prosodic grounding, causal coherence, and internal consistency using a 1–5 Likert scale. It is observed in the table below that VowelPrompt demonstrates significantly higher reasoning faithfulness than SpeechCueLLM across all four annotators. As shown in the table, SpeechCueLLM receives an average score of 3.14, while VowelPrompt receives an average score of 3.77, reflecting more accurate grounding in prosodic cues and greater internal coherence. These results confirm that VowelPrompt’s reasoning traces are not only more interpretable but also more consistently aligned with the prosodic evidence that drives its final predictions.
>
>
> | Methods                |   Evaluator 1  |   Evaluator 2  |    Evaluator 3  |  Evaluator 4  |  Average |
> | ---------------------- | :------: | :------: | :------: | :------: | :------: |
> | SpeechCueLLM           |   3.12   |   3.55   |   2.82   |   3.08   | 3.14     |
> | VowelPrompt (Ours) |4.05 | 3.96 | 3.42| 3.65| 3.77 |
>
> The results above are also added to Table 18 in Section A.10 of the revised paper.
>
>
> **Weakness 4. ”...is GRPO trading in-domain accuracy for distributional robustness, regularizing prosody usage, or mitigating spurious correlations? What hyperparameters or curricula drive this trade-off?”**
> **Question 4. ”GRPO dynamics: Can you explain or give learning curves and sensitivity analyses demonstrating when GRPO helps harm in-domain performance but helps cross-domain transfer?...”**
>
> The Knowledge Distillation (KL) weight in GRPO controls how strongly the policy is regularized toward the supervised SFT model, which effectively limits how far reinforcement learning can deviate from the source-domain distribution. To better understand the trade-off between the in-domain performance and the cross-domain performance, we conducted a sensitivity analysis by varying the KL weight and measuring both in-domain (IEMOCAP, MELD) and cross-domain (IEMOCAP→MELD, MELD→IEMOCAP) performance. As shown in the table below, decreasing the KL weight relaxes the constraint on the policy, resulting in slightly improved cross-domain robustness, indicating that the model relies less on dataset-specific lexical patterns and more on domain-invariant prosodic cues. On the other hand, increasing the KL weight leads to higher in-domain performance. Notably, both the in-domain and cross-domain performance of VowelPrompt vary only marginally across different values of the KL weight, which demonstrates that the GRPO-trained model is largely insensitive to the value of the KL weight. In addition, in the cross-domain setting, the GRPO data is from the source domain alone.
>
> |KLWeight|IEMOCAP|MELD|IEMOCAP→MELD|MELD→IEMOCAP|
> |--|:--:|:--:|:--:|:--:|
> |0.1|71.9|68.1|60.5|51.3|
> |0.25|73.4|69.6|60.2|51.7|
> |0.5|73.4|69.9|58.9|49.6|
> |1.0|73.6|70.0|58.4|49.2|
>
> The ablation study results above are also added to Table 19 in Section A.11 of the revised paper.
>
> **Weakness 5. ”... It is therefore unclear why an LLM is preferable to a purpose-built classifier over the same discrete features ...”**
> **Question 5. ”Value of LLMs: If the same phoneme-level attributes are available, what performance and interpretability gains remain when replacing the LLM with a standard classifier?...”**
>
> To demonstrate the necessity of an LLM-based architecture, we have conducted an ablation study comparing VowelPrompt against classifiers trained directly on the vowel-level prosodic features. In particular, the classifiers are a MLP classifier, a XGBoost, and a transformer-based classifier. In the transformer baseline, each phoneme is treated as a token, and its corresponding prosodic features are treated as the features of the token. The table below shows that VowelPrompt significantly outperforms all baseline classifiers, which demonstrates that access to the same attributes alone is insufficient. Vanilla classifiers fail to capture the contextual and linguistic dependencies that govern how vowel-level prosody conveys affect. In contrast, VowelPrompt leverages the LLM’s pretrained linguistic priors to integrate prosodic cues with lexical semantics, discourse context, and phonotactic patterns.
>
> | Datasets | XGBoost | MLP | Transformer| VowelPrompt |
> | --- | ---- | --- | --- | --- |
> | IEMOCAP        | 40.2      | 39.6  | 48.5    | 73.4      |
> | MELD              | 45.1      | 44.5   | 51.2    | 69.6      |
>
> The ablation study results above are also added to Table 20 in Section A.12 of the revised paper.

---

> ### Author Response · Authors · 2025-12-03
> **Response to the Official Review by Reviewer RfwA (Part 3)**
>
> **Weakness 6. ”...It is unclear how errors in forced alignment or noisy conditions propagate to final performance. Robustness analyses (e.g., perturbing boundaries, adding noise, cross-accent conditions) are limited.”**
> **Question 6. ”How robust is VowelPrompt to alignment errors and background noise? What is the performance degradation under realistic ASR/aligner noise or accented speech?”**
>
> To study the impact of incorrect vowel alignment, we have performed an ablation study that perturbed 5%, 10%, and 15% of the boundaries of the vowels in the alignment results. The study is performed on MELD using LLaMA-3-8B-Instruct. In particular, for each selected vowel segment, we randomly shifted its start or end times by 50% of its original duration. It is observed in the table below that the performance of VowelPrompt is robust to the perturbation of the boundaries of the vowels in the alignment results and consistently achieves significantly better performance than SpeechCueLLM. For example, even with 15% of the vowel boundaries perturbed, VowelPrompt still achieves a Weighted F1 of 69.11%, which outperforms SpeechCueLLM by 2.04%.
>
> | Method | Perturbation Ratio | Weighted F1 |
> | --- | --- | --- |
> | SpeechCueLLM |0 | 67.07 %  |
> | VowelPrompt  |0 | 69.61 %  |
> | VowelPrompt  |5% |  69.50 %|
> | VowelPrompt  |10% |  69.23 % |
> | VowelPrompt  |15% |  69.11 % |
>
> Moreover, we have evaluated the performance of ASVP-ESD on a heavily accented dataset, ASVP-ESD [5], in Section 4.5. The results show that VowelPrompt significantly outperforms all competing baseline methods on ASVP-ESD.
>
> The ablation study results above are also added to Table 21 in Section A.13 of the revised paper.
>
> **References**
>
> [1] Jongman, Allard. "Phonetics of fricatives." Oxford Research Encyclopedia of Linguistics. 2024.
>
> [2] Diehl, Randy L. "Acoustic and auditory phonetics: the adaptive design of speech sound systems." Philosophical Transactions of the Royal Society B: Biological Sciences 363.1493 (2008): 965-978.
>
> [3] Hsu, Wei-Ning, et al. "Hubert: Self-supervised speech representation learning by masked prediction of hidden units." IEEE/ACM transactions on audio, speech, and language processing 29 (2021): 3451-3460.
>
> [4] Baevski, Alexei, et al. "wav2vec 2.0: A framework for self-supervised learning of speech representations." Advances in neural information processing systems 33 (2020): 12449-12460.
>
> [5] Landry Dejoli Tientcheu Touko, Qianhua He, and Wei Xie. Audio, speech and vision processing lab emotional sound database (asvp-esd). Dataset, May 2021. Audio, Speech and Vision Processing Lab.

---

### Official Review · Reviewer_ksVz · 2025-11-04

**Soundness:** 3
**Presentation:** 3
**Contribution:** 2
**Rating:** 8
**Confidence:** 3

**Summary:**

The paper proposes a novel Speech Emotion Recognition approach that augments speech transcripts with vowel-level prosodic descriptors (F0 level/slope/variation, intensity level/variation, duration) extracted via forced alignment. These descriptors are then used in an LLM-as-classifier framework, fine-tuned via supervised fine-tuning (SFT) and reinforcement learning (RLVR) from GPT-4o (LLM as oracle) for reasoning traces.

To account for cross-speaker and vowel variability, the authors first perform speaker- and vowel-type normalization, then quantile-bin the prosodic statistics and convert them into natural-language tokens, which are appended to the transcript so the LLM can reason over lexical and localized prosody. The method is evaluated on IEMOCAP, MELD, CaFE, EmoDB, and ASVP-ESD (mixed-lingual), showing consistent gains over transcript-only and sentence-level prosody prompts, with interpretable rationales.

**Strengths:**

The proposed method has several strengths, which can be grouped into two categories:

**Conceptual**: the method provides a privacy-oriented, interpretable formulation of SER by augmenting transcripts with symbolic, vowel-level prosody tokens (F0 level, intensity, duration), allowing emotion inference without raw audio at inference. This design is linguistically grounded (vowels are typically stable prosodic carriers), separates lexical content from paralinguistic cues, and supports closed-set classification, addressing both explainability and data-minimization goals.

**Implementation strengths**: the end-to-end pipeline is clear and modular: forced alignment, followed by speaker and vowel normalization, then quantile binning, conversion to natural-language tokens, and an LLM-as-classifier stage. Each step is inspectable and easily ablated. Training uses a practical two-stage recipe (supervised fine-tuning followed by RLVR/GRPO with verifiable rewards) and performs consistently across datasets and model families, improving over transcript-only and sentence-level prosody prompts. At deployment, the approach seems easy to integrate with existing text infrastructures.

**Weaknesses:**

Because the oracle used for RLVR traces (GPT-4o) is itself an LLM, there is a material risk that fine-tuned student LLMs learn spurious lexical or formatting heuristics unrelated to the intended prosodic mechanism. To establish that the model uses vowel-level prosodic tokens rather than incidental cues, please include controlled counterfactual ablations that preserve input statistics while breaking the hypothesized channel. Some examples to test:
- Transcript shuffle control: randomly permute word order, or replace content words with frequency-matched synonyms, while keeping the vowel-prosody tokens intact. Performance should remain near the original if prosody carries the signal and should drop sharply if lexical priors dominate.
- Prosody permutation control: permute the vowel-prosody descriptors across utterances within a mini-batch while keeping transcripts fixed. Performance should drop to near chance if the model relies on the prosodic channel.
- Matched-marginal placebo: replace prosody tokens with random draws from their empirical per-vowel distribution so that marginals are preserved but alignment is broken. This controls for token frequency or style leakage.
- Cross-swap (counterfactual consistency): for the same transcript, attach prosody from an utterance of a different emotion. Predicted labels should flip in the direction implied by the swapped prosody if the mechanism is causal.

Also, the paper does not analyze how the discrete labels are tokenized by each model’s tokenizer. If some labels map to a single token while others split into multi-token sequences, the model may exhibit unequal calibration and decoding bias unrelated to prosodic evidence. An ablation study can include:

- Tokenization audit per model: for each model used (for example, LLaMA-3-8B, Qwen-2-7B), report (i) tokenization of each label verbalizer (angry, sad, neutral, happy, excited), including the number of tokens and subword pieces, and (ii) empirical prior token likelihoods for these verbalizers under pseudo-neutral prompts, for example, a transcript without prosody information.
- Label set distribution: compute normalized label probabilities via the product of subtoken log-probabilities for multi-token labels and report calibration. This assesses whether the decoding process favors specific verbalizers independent of evidence.
- Label name permutation: assign the emotion categories to some random verbalizer strings to test sensitivity to label names. If the model captures the underlying categories rather than specific strings, performance should remain stable.

Answering some of these questions would strengthen the conceptual foundation of the paper and the interpretability of the suggested method.

**Questions:**

Some follow-up questions:

- Are `Speaker_{%d}` tokens used only as placeholders to distinguish multiple speakers within an utterance, or do they carry any stable identity across utterances and speakers?

- Related work organization: The “Vowel-Centric Prosody in Emotional Speech” paragraph and the first paragraph of Section 3.1 can be merged. This would reduce redundancy and improve flow by introducing the vowel-centric rationale once, then immediately presenting the operational relevance.

- Cross-linguistic normalization and scope: the manuscript states: ``We ensure cross-linguistic consistency and compatibility with multilingual phonetic analysis pipelines …`` and later ``To control for cross-lingual variation in prosodic realization, we further perform normalization at the language level. For each language, we compute global means and standard deviations for each prosodic feature and apply z-score normalization within that language``...  Please specify the language set to which these claims apply. Cross-linguistic prosody-emotion mappings differ across languages, especially in tone languages such as Vietnamese and Chinese where pitch contours are lexical. It would be helpful to limit claims to the languages actually evaluated (for example, English, French, German).

- Determinism of “parameter-free” processing and oracle traces: The manuscript states that the process is ``parameter-free, ensuring transparency and reproducibility,`` and that ``gold reasoning traces [are] automatically generated by a high-capacity text-only LLM such as GPT-4o.``... Are these oracle traces generated deterministically. Please report the decoding settings used for GPT-4o, including temperature, top-p, and any seed control. If temperature was set to zero, state this clearly. If not, either provide information about variance across repeated generations and whether multiple samples were filtered or selected, or elaborate on the "reproducibility" claim.

- RLVR objective and the role of reasoning traces: on the RLVR stage, how are the reasoning traces used. Are rationales part of the reward signal beyond format verification? Please clarify whether any rationale consistency checks are part of the verifiable objectives.

---

> ### Author Response · Authors · 2025-12-03
> **Response to the Official Review by Reviewer ksVz (Part 1)**
>
> We appreciate the review and the suggestions in this review. The raised issues are addressed below.
>
> **Responses to the Weaknesses**
>
> **1. ”...To establish that the model uses vowel-level prosodic tokens rather than incidental cues, please include controlled counterfactual ablations that preserve input statistics while breaking the hypothesized channel. Some examples to test: Transcript shuffle control…Prosody permutation control…Matched-marginal placebo….Cross-swap (counterfactual consistency)...
> ”**
>
> To demonstrate that VowelPrompt relies on vowel-level prosodic descriptors instead of spurious lexical or formatting heuristics inherited from the oracle reasoning traces, we conducted a series of ablation studies on transcript shuffle control, prosody permutation control, matched-marginal placebo, and cross-swap. The ablation study is performed on MELD using LLaMA-3-8B-Instruct trained with GRPO.
> In the study on transcript shuffle control, we randomly permute the word order while keeping the vowel-level prosodic descriptors intact. It is observed in the table below that the performance of VowelPrompt only marginally decreases under this perturbation, indicating that lexical ordering or content identity is not the dominant predictive signal, and the prediction of VowelPrompt heavily relies on the vowel-level prosodic information coupled with the lexical information of the vowels.
> In the study on prosody permutation control, we permute the vowel-prosody descriptors across utterances within each training mini-batch while leaving transcripts unchanged.  It is observed in the table below that the prosody permutation leads to a significant performance degradation, which demonstrates that VowelPrompt significantly depends on the alignment between vowel-level prosodic cues and the corresponding utterances, instead of relying on the transcript alone.
> We further perform a matched-marginal placebo experiment, where prosody tokens are replaced with random draws from their empirical per-vowel distributions. This preserves the marginal statistics, token frequencies, and style patterns but destroys semantic grounding. It is observed in the table below that the performance of the ablation model decreases significantly, which demonstrates that VowelPrompt does not rely on superficial token regularities and instead requires aligned prosodic descriptors to make accurate predictions.
>
>
> | Methods                         | Weighted F1 (%) |
> |---------------------------------|:--------------:|
> | VowelPrompt (Prosody Permutation)             | 41.7          |
> | VowelPrompt (Matched-Marginal Placebo)        | 44.1          |
> | VowelPrompt (Transcript Shuffle)              | 67.0      |
> | VowelPrompt         | 68.9     |
>
> Finally, we perform a cross-swap counterfactual consistency experiment, where we preserve the transcript but attach prosodic descriptors extracted from utterances belonging to a different emotion category. The study is performed on the happy and the sad emotions in IEMOCAP. It is observed that the predicted emotion systematically follows the swapped prosodic profile rather than the lexical content. As shown in the table below, when happy utterances are paired with sad prosody, the proportion of predictions labeled as sad increases from 18.7% to 45.8%, while retaining the original transcript. Conversely, when sad utterances are paired with happy prosody, the proportion of happy predictions increases from 27.5% to 51.0%. The above results demonstrate that VowelPrompt does not merely memorize lexical patterns but actively attributes emotional prediction to vowel-level prosodic cues, which provides direct causal evidence that the prosodic descriptors, rather than text alone, drive the model’s decision-making.
>
> | Ground-Truth Emotion | Prosody Source | Predicted Happy (%) | Predicted Sad (%) |
> |---------------------|---------------|:-------------------:|:-----------------:|
> | Happy               | Happy         | 81.3            | 18.7              |
> | Happy               | Sad           | 54.2                | 45.8          |
> | Sad                 | Sad           | 27.5                | 72.5          |
> | Sad                 | Happy         |51.0           | 49.0              |
>
> The ablation study results above are also added to Table 13 and Table 14 in Section A.6 of the revised paper.

---

> > ### Author Response · Authors · 2025-12-03
> > **Response to the Official Review by Reviewer ksVz (Part 2)**
> >
> > **2. ”Also, the paper does not analyze how the discrete labels are tokenized by each model’s tokenizer... the model may exhibit unequal calibration and decoding bias unrelated to prosodic evidence. ….”**
> >
> > To study the impact of the tokenization behavior of the labels, we first perform a study on the tokenization behavior of the IEMOCAP verbalizers, including angry, excited, happy, neutral, and sad, under both the LLaMA-3-8B and Qwen-2-7B tokenizers. In both models, happy, neutral, and sad are each encoded as single-token verbalizers, while angry (['ang','ry']) and excited (['exc', 'ited']) are consistently split into two subword units.
> > To study the impact of the decoding bias arising from such variation, we replaced all emotion labels with synthetic two-letter tokens (happy→ha, sad→sa, angry→an, neutral→ne, excited→ex) that are uniformly represented as single tokens across both tokenizers. We then randomly permuted the emotion verbalizer mapping 10 times, thereby eliminating any lexical or semantic prior that the tokenizer could exploit.
> > The results show that replacing the original emotion verbalizers with synthetic two-letter tokens leads to only a marginal performance drop on VowelPrompt, which demonstrates that VowelPrompt is marginally impacted by the decoding bias. Notably, the impact is significantly smaller for VowelPrompt (73.0% → 71.7% → 71.0%) compared to SpeechCueLLM (71.5% → 67.8% → 65.2%), which demonstrates that VowelPrompt is significantly more robust to label perturbations because the predictions are grounded in detailed vowel-level prosodic cues rather than lexical or tokenization-based priors associated with the verbalizers.
> >
> > |Methods|WeightedF1(%)|
> > |--|--|
> > |VowelPrompt|73.0|
> > |SpeechCueLLM|71.5|
> > |VowelPrompt (Two-Letter)|71.7|
> > |SpeechCueLLM (Two-Letter)|67.8|
> > |VowelPrompt (Two-LetterPermutated)|71.0|
> > |SpeechCueLLM (Two-LetterPermutated)|65.2|
> >
> > The ablation study results above are also added to Table 15 in Section A.7 of the revised paper.
> >
> > ** Responses to the Questions**
> >
> > **1. ”Are Speaker_{%d} tokens used only as placeholders…”**
> >
> > The Speaker_{%d} tokens are used solely as local identifiers within each conversation to distinguish different speakers appearing in the prompt following the settings in existing work [1]. These tokens do not encode persistent speaker identities across utterances, sessions, or datasets.
> >
> > **2. ”Related work organization: The “Vowel-Centric Prosody in Emotional Speech” paragraph and the first paragraph of Section 3.1 can be merged...”**
> >
> > We have merged the first paragraph of Section 3.1 into the paragraph on “Vowel-Centric Prosody in Emotional Speech” in the section on related works.
> >
> > **3. ”Cross-linguistic normalization and scope: the manuscript states: We ensure cross-linguistic consistency and compatibility with multilingual phonetic analysis pipelines ... It would be helpful to limit claims to the languages actually evaluated (for example, English, French, German).”**
> >
> > In Section 3.3 of the revised paper, we have now explicitly restricted the scope of our cross-linguistic normalization claim to the languages actually evaluated in this work, namely English, German, and French.
> >
> > **4. ”...gold reasoning traces [are] automatically generated by a high-capacity text-only LLM such as GPT-4o.... Are these oracle traces generated deterministically…”**
> >
> > The gold reasoning traces used in the SFT stage were generated deterministically using GPT-4o with temperature = 0.0 and top-p = 1.0, and no sampling seeds or stochastic decoding components were applied. Under such settings, GPT-4o performs greedy decoding, which ensures that each input prompt yields a consistent output across repeated generations with no variance.
> >
> > **5. ”RLVR objective and the role of reasoning traces: on the RLVR stage, how are the reasoning traces used... Please clarify whether any rationale consistency checks are part of the verifiable objectives.”**
> >
> > In the RLVR stage, the reasoning traces are not directly optimized for semantic correctness or rationale quality. Instead, they function as an explicit structural component that separates the model’s internal reasoning from its final emotion prediction. Inspired by DeepSeek-R1 [2], the reward used in RLVR consists of two verifiable elements. The first evaluates whether the predicted label matches the ground-truth emotion category, and the second verifies that the model output conforms to the required structure by including syntactically valid <think>...</think> and <answer>...</answer> blocks. No additional semantic checks, ranking mechanisms, or rationale-quality assessments are applied to the reasoning traces.
> >
> > **References**
> >
> > [1] Wu, Zehui, et al. "Beyond silent letters: Amplifying llms in emotion recognition with vocal nuances." NAACL Findings 2025.
> >
> > [2] Guo, Daya, et al. "Deepseek-r1 incentivizes reasoning in llms through reinforcement learning." Nature 645.8081 (2025): 633-638.

---

### Author Response · Authors · 2025-12-03
**Summary of Revision (Part 1)**

Dear AC,

Thank you for your time handling and reviewing this paper. We have thoroughly addressed all reviewers’ concerns, which is also reflected in the revised paper. Below is a concise summary of our key clarifications and revisions addressing all the concerns.

**1. Summary of Revisions Addressing the Comments by Reviewer RfwA**

- **Reviewer RfwA is concerned that the exclusion of the consonant-level cues is not justified.**
To address this concern, we first clarify that our design focuses on vowels because they provide the most reliable and acoustically stable carriers of prosodic variation. In contrast, consonantal cues are brief, low-intensity, and spectrally unstable. Furthermore, we have performed an ablation study comparing the effectiveness of using vowel-only, consonant-only, and all-phoneme cues on datasets of different languages. The results show that using vowel-level cues alone achieves significantly better performance than using consonant-level cues alone. Incorporating both vowel- and consonant-level cues improves the performance of VowelPrompt on German.

- **Reviewer RfwA is concerned that the non-LLM SER baseline methods are not compared.**
To address this concern, we have compared VowelPrompt with non-LLM speech emotion recognition baselines using state-of-the-art self-supervised speech models, including HuBERT-large and wav2vec2-large. The results show that VowelPrompt consistently outperforms HuBERT-large and wav2vec2-large in both in-domain and cross-domain speech emotion recognition.

- **Reviewer RfwA is concerned that the reasoning faithfulness of VowelPrompt is not evaluated.**
To address this concern, we conducted a human evaluation of the faithfulness of the reasoning traces generated by VowelPrompt with respect to prosodic grounding, causal coherence, and internal consistency. The results show that the reasoning traces generated by VowelPrompt are more consistently aligned with the prosodic evidence that drives its final predictions compared to the baseline method.

- **Reviewer RfwA is concerned that the trade-off between the in-domain and cross-domain performance of VowelPrompt trained by GRPO is not discussed.**
To address this concern, we have clarified that the Knowledge Distillation (KL) weight in GRPO controls the trade-off between the in-domain and cross-domain performance. Higher KL weight usually leads to better in-domain performance, while lower KL weight usually leads to better cross-domain performance. Furthermore, we have performed a sensitivity analysis showing that although such a trade-off also applies to VowelPrompt, the performance of VowelPrompt is not sensitive to the value of the KL weight.

- **Reviewer RfwA is concerned that VowelPrompt is not compared with simple feature-to-label classifiers, such as gradient-boosted tree, MLP, or Transformer, to justify the use of LLM.**
To address this concern, we have compared VowelPrompt with baseline classifiers built with MLP, XGBoost, and transformer. The results show that VowelPrompt significantly outperforms all baseline classifiers, which is because vanilla classifiers fail to capture the contextual and linguistic dependencies that govern how vowel-level prosody conveys affect. In contrast, VowelPrompt leverages the LLM’s pretrained linguistic priors to integrate prosodic cues with lexical semantics, discourse context, and phonotactic patterns.

- **Reviewer RfwA is concerned about the robustness of VowelPrompt to phoneme alignment errors and noises.**
To address this concern, we have performed an experiment that perturbs the vowel alignment results used for VowelPrompt. The results show that the performance of VowelPrompt is robust to the alignment noise. Furthermore, Table 7 in Section 4.5 of our paper shows that VowelPrompt significantly outperforms all competing baseline methods on a heavily accented dataset, ASVP-ESD.

**2. Summary of Revisions Addressing the Comments by Reviewer W37J**

- **Reviewer W37J is concerned about the public availability of the tools used to obtain the prosodic descriptors.**
We have clarified that prosodic descriptors in our framework are generated using publicly available tools and resources, including Montreal Forced Aligner (MFA) [1], International Phonetic Alphabet (IPA) [2], and Praat [3].

- **Reviewer W37J is concerned that the vowel boundaries obtained from the forced alignment may be uncertain and inaccurate.**
To address this concern, we have performed an experiment that perturbs the vowel alignment results used for VowelPrompt. The results show that the performance of VowelPrompt is robust to the alignment noise.

- **Reviewer W37J is concerned about the impact of the speech rate on the performance of VowelPrompt.**
To address this concern, we have performed an ablation study by categorizing testing utterances according to their speech rate. The results show that VowelPrompt consistently outperforms the competing baseline methods across all speech rate ranges.

---

> ### Author Response · Authors · 2025-12-03
> **Summary of Revision (Part 2)**
>
> - **Reviewer W37J asked if vowel level means at the individual distinct vowel level or by vowel-groups.**
> We have clarified that vowel level refers to individual vowel segments instead of coarse-grained vowel groupings.
>
> - **Reviewer W37J asked how the difference in categories was accounted for during the cross-domain evaluation.**
> We have clarified that a unified label space is built by mapping semantically equivalent categories following [4]. Moreover, the evaluation prompt explicitly enumerates the emotion categories available in the target domain.
>
> - **Reviewer W37J asked if all emotion categories of the datasets in Table 2 are used.**
> We have clarified that all the emotion categories are used.
>
> **3. Summary of Revisions Addressing the Comments by Reviewer b43y**
>
> - **Reviewer b43y is concerned that the idea of appending prosody descriptions is not novel, and the novelty of VowelPrompt lies in expanding it to the vowel-level.**
> To address this concern, we have clarified that the contribution of VowelPrompt is not only the addition of prosodic descriptors to the input, but also the introduction of a reasoning-centric framework that enables the model to perform structured inference over fine-grained, vowel-level prosodic cues. VowelPrompt transforms prosodic information into phoneme-aligned fine-grained descriptors that are explicitly invoked within the model’s internal reasoning process. We have performed an ablation study showing that VowelPrompt consistently outperforms SpeechCueLLM, with and without reasoning enabled, demonstrating that the improvements stem from the model’s ability to causally integrate and reason over the fine-grained cues. Moreover, the benefits of such fine-grained reasoning capability are even more significant under the cross-domain transfer setting, as evidenced in Table 5 of our paper.
>
> - **Reviewer b43y is concerned that the comparisons between the proposed model and the baselines are not conducted under both conditions, with reasoning enabled for all methods or with reasoning disabled for all methods.**
> To address this concern, we have performed an ablation study by enabling and disabling reasoning/thinking in both the SFT and the GRPO settings. The results show that VowelPrompt consistently outperforms the competing baseline methods, demonstrating that the advantages of VowelPrompt stem from both the fine-grained vowel-level prosodic cues and the reasoning capability.
>
> - **Reviewer b43y is concerned that the GRPO does not improve the in-domain performance of VowelPrompt.**
> To address this concern, we have clarified that GRPO provides no additional supervisory signal beyond the ground-truth label, which is already used in the SFT setting. When reasoning is enabled, GRPO introduces a verifiable reward that incentivizes reasoning. This reinforcement signal induces more stable and systematic decision-making, which is significantly effective under a distributional shift, as evidenced by the cross-domain results in Table 5 of our paper. Such observations are also consistent with recent analyses demonstrating that SFT largely determines in-distribution performance, while RLVR with GRPO contributes to robustness beyond the training distribution [5].
>
> We have also addressed the issues from Reviewer ksVz in our rebuttals and the revised paper.
>
> **References**
>
> [1] McAuliffe, Michael, et al. "Montreal forced aligner: Trainable text-speech alignment using kaldi." Interspeech. Vol. 2017. 2017.
>
> [2] International Phonetic Association. Handbook of the International Phonetic Association: A guide to the use of the International Phonetic Alphabet. Cambridge University Press, 1999.
>
> [3] Paul Boersma and David Weenink. Praat, a system for doing phonetics by computer. Glot International, 5(9/10):341–345, 2001.
>
> [4] Lei, Shanglin, et al. "Instructerc: Reforming emotion recognition in conversation with multi-task retrieval-augmented large language models." arXiv preprint arXiv:2309.11911 (2023).
>
> [5] Kirk, Robert, et al. "Understanding the Effects of RLHF on LLM Generalisation and Diversity." ICLR 2024.

---

### Meta-Review · Area_Chair_BqQw · 2026-01-07

**Summary:**

This paper presents an approach to improve LLM-based emotion recognition through the use of vowel-level prosodic cues with the semantics of the textual content. These cues include pitch, energy, and durations of vowel segments, converted into natural language descriptions, enabling joint reasoning over text and prosodic content. The work uses supervised fine-tuning and reinforcement learning (e.g., GRPO) and demonstrate4s improved results and interpretability in zero-shot, fine-tuned, cross-domain, and cross-lingual setups. The approach is evaluated on several benchmark datasets: IEMOCAP, MELD, CaFE, EmoDB, and ASVP-ESD, and results in improvements over using only the text content.

**Reviewer Concerns:**

Reviewers ksVz and RfwA requested several detailed analysis experiments, which were all included during the rebuttal period. I believe there are no outstanding reviewer concerns.

**Reviewer Scores:**

I expect all reviewers would increase their ratings (although none of them responded during the discussions period), given that the rebuttals addressed each of the weaknesses.

---

### Decision · Program_Chairs · 2026-01-26

Accept (Poster)